# IMPROVED DIFFUSION-BASED GENERATIVE MODEL WITH BETTER ADVERSARIAL ROBUSTNESS

**Zekun Wang**[*1], **Mingyang Yi**[*†2], **Shuchen Xue**[3,4], **Zhenguo Li**[5], **Ming Liu**[†1], **Bing Qin**[1], **Zhi-Ming Ma**[3,4]

[1]Harbin Institute of Technology    [2]Renmin University of China
[3]Academy of Mathematics and Systems Science, Chinese Academy of Sciences
[4]University of Chinese Academy of Sciences    [5]Huawei Noah's Ark Lab
`zkwang@ir.hit.edu.cn yimingyang@ruc.edu.cn`
`xueshuchen17@mails.ucas.ac.cn`

## ABSTRACT

Diffusion Probabilistic Models (DPMs) have achieved significant success in generative tasks. However, their training and sampling processes suffer from the issue of distribution mismatch. During the denoising process, the input data distributions differ between the training and inference stages, potentially leading to inaccurate data generation. To obviate this, we analyze the training objective of DPMs and theoretically demonstrate that this mismatch can be alleviated through Distributionally Robust Optimization (DRO), which is equivalent to performing robustness-driven Adversarial Training (AT) on DPMs. Furthermore, for the recently proposed Consistency Model (CM), which distills the inference process of the DPM, we prove that its training objective also encounters the mismatch issue. Fortunately, this issue can be mitigated by AT as well. Based on these insights, we propose to conduct efficient AT on both DPM and CM. Finally, extensive empirical studies validate the effectiveness of AT in diffusion-based models. The code is available at `https://github.com/kugwzk/AT_Diff`.

## 1 INTRODUCTION

Diffusion Probabilistic Models (DPMs) (Ho et al., 2020; Song et al., 2020; Yi et al., 2024) have achieved remarkable success across a wide range of generative tasks such as image synthesis (Dhariwal & Nichol, 2021; Rombach et al., 2022; Ho et al., 2022a), video generation (Ho et al., 2022b; Blattmann et al., 2023), text-to-image generation (Nichol et al.; Ramesh et al., 2022; Saharia et al., 2022), *etc*. The core mechanism of DPMs involves a forward diffusion process that progressively injects noise into the data, followed by a reverse process that learns to generate data by denoising the noise. Unlike traditional generative models such as GANs(Goodfellow et al., 2014) or VAEs (Kingma & Welling, 2013), which directly map an easily sampled latent variable (e.g., Gaussian noise) to the target data through a single network function evaluation (NFE), DPMs adopt a gradual denoising approach that requires multiple NFEs (Song et al., 2022; Salimans & Ho, 2022; Lu et al., 2022b; Ma et al., 2024). However, this noising-then-denoising process introduces a distribution mismatch between the training and sampling stages, potentially leading to inaccuracies in the generated outputs.

Concretely, during the training stage, the model is learned to predict the noise in ground-truth noisy data derived from the training set. In contrast, during the inference stage, the input distribution is obtained from the output generated by the DPM in the previous step, which differs from the training phase, caused by the inaccurate estimation of the score function due to training (Song et al., 2021; Yi et al., 2023a) and the discretization error (Chen et al., 2022; Li et al., 2023; Xue et al., 2024b;a) brought by sampling. Such distribution mismatches are referred to as *Exposure Bias*, which has been discussed in auto-regressive language models (Bengio et al., 2015; Ranzato et al., 2016).

Recently, the aforementioned distribution mismatch problem in diffusion has been also recognized by (Ning et al., 2023; Li & van der Schaar, 2024; Ren et al., 2024; Ning et al., 2024; Li et al.,

---

[*]Equal contribution
[†]Corresponding author

2024; Lou & Ermon, 2023). However, these studies are either rely on strong mismatch distributional assumptions (e.g., Gaussian) (Ning et al., 2023; 2024; Ren et al., 2024) or incur significant additional computational costs (Li & van der Schaar, 2024). This indicates that a more practical solution to this problem has been overlooked until now. To bridge this gap, we begin with the discrete DPM introduced in (Ho et al., 2020). Intuitively, although there is a mismatch between training and inference, the distributions of intermediate noise generated during the inference stage are close to the ground-truth distributions observed during training. Therefore, improving the distributional robustness (Yi et al., 2021; Namkoong, 2019; Shapiro, 2017) (which measures the robustness of the model to distributional perturbations in training data) of DPM mitigates the distribution mismatch problem. To achieve this, we refer to Distribution Robust Optimization (DRO) (Shapiro, 2017; Namkoong, 2019), which aims to improve the distributional robustness of models. We then prove that applying DRO to DPM is mathematically equivalent to implementing *robustness-driven* Adversarial Training (AT) (Madry et al., 2018; Shafahi et al., 2019; Yi et al., 2021) on DPM. [1] Following the DRO framework, we also analyze the recently proposed diffusion-based Consistency Model (CM) (Song et al., 2023; Luo et al., 2023) which distills the trajectory of DPM into a model with one NFE generation. We first prove that the training objective of CM similarly suffers from the mismatch issue as in multi-step DPM. Moreover, the issue can also be mitigated by implementing AT. Therefore, for both DPM and CM, we propose to apply efficient AT (e.g., "Free-AT" (Shafahi et al., 2019)) during their training stages to mitigate the distribution mismatch problem.[2] Finally, we summarize our contributions as follows.

- We conduct an in-depth analysis of the diffusion-based models (DPM and CM) from a theoretical perspective and systematically characterize its distribution mismatch problem.
- For both DPM and CM, we theoretically show that their mismatch problem is mitigated by DRO, which is equivalent to implementing AT with proved error bounds during training.
- We propose to conduct efficient AT on both DPM and CM in various tasks, including image generation on `CIFAR10` 32×32(Krizhevsky & Hinton, 2009) and `ImageNet` 64×64 (Deng et al., 2009), and zero-shot Text-to-Image (T2I) generation on MS-COCO 512×512 (Lin et al., 2014b). Extensive experimental results illustrate the effectiveness of the proposed AT training method in alleviating the distribution mismatch of DPM and CM.

## 2 RELATED WORK

**Distribution Mismatch in DPM.** The problem is analogous to the exposure bias in auto-regressive language models (Bengio et al., 2015; Ranzato et al., 2016; Shen et al., 2016; Rennie et al., 2017; Zhang et al., 2019c), whereas the next word prediction (Radford et al., 2019) relies on tokens predicted by the model in the inference stage, which may be mismatched with the ground-truth one taken in the training stage. The similarity to DPMs becomes evident due to their gradual denoising generation process. Ning et al. (2023) and Ning et al. (2024) propose adding extra Gaussian perturbation during the training stage or data-dependent perturbation during the inference stage, to mitigate this issue. Following this line of work, several methods are further proposed. For instance, to reduce the accumulated discrepancy between the intermediate noisy data in the training and inference stages, Li et al. (2024) search for a suboptimal mismatched input time step of the model to conduct inference. Similarly, Li & van der Schaar (2024) and Ren et al. (2024) directly minimize the difference between the generated intermediate noisy data and the ground-truth data. However, these methods either rely on strong assumptions (Ning et al., 2023; 2024; Li et al., 2024; Ren et al., 2024) or are computationally expensive (Li & van der Schaar, 2024). In contrast, we are the first to explore the distribution mismatch problem from the perspective of DRO. Meanwhile, our proposed AT with strong theoretical foundations is both simple and efficient, compared with the existing methods.

**Adversarial Training and DRO.** In this paper, we leverage the Distributionally Robust Optimization (DRO) (Shapiro, 2017; Namkoong, 2019; Yi et al., 2021; Sinha et al., 2018; Wang et al., 2022; Yi et al., 2023b) to improve the distributional robustness of DPM and CM, thereby mitigating

---

[1]Note that the "adversarial" here refers to perturbation to input training data, instead of the adversarial of generator-discriminator in GAN (Goodfellow et al., 2014).

[2]Notably, the standard AT (Madry et al., 2018) solves a minimax problem that slows the training process. The efficient AT has no extra computational cost compared to the standard training ones (Shafahi et al., 2019).

the distribution mismatch problem. As demonstrated in (Sinha et al., 2018; Yi et al., 2021; Lee & Raginsky, 2018), we link the DRO with AT (Madry et al., 2018; Goodfellow et al., 2015), which is designed to improve the input (instead of distributional) robustness of the model. In supervised learning, the adversarial examples generated by efficient AT methods (Shafahi et al., 2019; Zhang et al., 2019a;b; Zhu et al., 2020; Jiang et al., 2020) have been proven to be efficient augmented data to improve the robustness and generalization performance of models (Rebuffi et al., 2021; Wu et al., 2020; Yi et al., 2021). In this paper, we further verify that the AT generated adversarial augmented examples are also beneficial for generative models DPM and CM.

In addition, recent studies (Nie et al., 2022; Wang et al., 2023; Zhang et al., 2023) utilize DPM to generate examples in adversarial training to improve the robustness of the classification model. This is quite different from the method in this paper, as we focus on employing AT during training of diffusion-based model to improve its distributional robustness to alleviate the distribution mismatching.

## 3 PRELIMINARY

**Diffusion Probabilistic Models.** DPM (Sohl-Dickstein et al., 2015; Ho et al., 2020) constructs the Markov chain $\boldsymbol{x}_t$ by transition kernel $q(\boldsymbol{x}_{t+1} \mid \boldsymbol{x}_t) = \mathcal{N}(\sqrt{\alpha_{t+1}}\boldsymbol{x}_t, (1-\alpha_{t+1})\boldsymbol{I})$, where $\alpha_1, \cdots, \alpha_T$ are in $[0, 1]$. Let $\bar{\alpha}_t := \Pi_{s=1}^t \alpha_s$, and $\boldsymbol{x}_0 \sim q$ be ground-truth data. Then, for $\boldsymbol{x}_t$, it holds

$$\boldsymbol{x}_t = \sqrt{\bar{\alpha}_t}\boldsymbol{x}_0 + \sqrt{1 - \bar{\alpha}_t}\boldsymbol{\epsilon}_t \qquad t = 1, \cdots, T, \tag{1}$$

with $\boldsymbol{\epsilon}_t \sim \mathcal{N}(0, \boldsymbol{I})$. The reverse process $p_{\boldsymbol{\theta}}(\boldsymbol{x}_t \mid \boldsymbol{x}_{t+1})$ is parameterized as

$$p_{\boldsymbol{\theta}}(\boldsymbol{x}_t \mid \boldsymbol{x}_{t+1}) = \mathcal{N}(\mu_{\boldsymbol{\theta}}(\boldsymbol{x}_{t+1}, t+1), \sigma_{t+1}^2 \boldsymbol{I}), \tag{2}$$

where $\sigma_{t+1}^2 = 1 - \alpha_{t+1}$. To learn $p_{\boldsymbol{\theta}}(\boldsymbol{x}_t \mid \boldsymbol{x}_{t+1})$, a standard method is to minimize the following evidence lower bound of negative log-likelihood (NLL) (Ho et al., 2020),

$$-\mathbb{E}_q[\log p_{\boldsymbol{\theta}}(\boldsymbol{x}_0)] \leq \mathbb{E}_q\left[-\log \frac{p_{\boldsymbol{\theta}}(\boldsymbol{x}_{0:T})}{q(\boldsymbol{x}_{1:T} \mid \boldsymbol{x}_0)}\right]. \tag{3}$$

Here, minimizing the ELBO in the r.h.s. of above inequality links to $p_{\boldsymbol{\theta}}(\boldsymbol{x}_t \mid \boldsymbol{x}_{t+1})$ since it is equivalent to minimizing the following rewritten objective

$$\min_{\boldsymbol{\theta}} \left\{ D_{KL}(q(\boldsymbol{x}_T) \parallel p_{\boldsymbol{\theta}}(\boldsymbol{x}_T)) + \sum_{t=0}^{T-1} \underbrace{D_{KL}(q(\boldsymbol{x}_t \mid \boldsymbol{x}_{t+1}) \parallel p_{\boldsymbol{\theta}}(\boldsymbol{x}_t \mid \boldsymbol{x}_{t+1}))}_{L_t} \right\}, \tag{4}$$

as in (Ho et al., 2020; Bao et al., 2022; Yi et al., 2023a). Here, the conditional Kullback–Leibler (KL) divergence $D_{KL}(q(\boldsymbol{x}_t \mid \boldsymbol{x}_{t+1}) \parallel p(\boldsymbol{x}_t \mid \boldsymbol{x}_{t+1})) = \int q(\boldsymbol{x}_t \mid \boldsymbol{x}_{t+1}) \log \frac{q(\boldsymbol{x}_t \mid \boldsymbol{x}_{t+1})}{p(\boldsymbol{x}_t \mid \boldsymbol{x}_{t+1})} d\boldsymbol{x}_t d\boldsymbol{x}_{t+1}$ (Duchi, 2016), and minimizing $L_t$ is equivalent to solve the following noise prediction problem

$$\min_{\boldsymbol{\theta}} \mathbb{E}\left[\left\|\boldsymbol{\epsilon}_{\boldsymbol{\theta}}(\sqrt{\bar{\alpha}_t}\boldsymbol{x}_0 + \sqrt{1 - \bar{\alpha}_t}\boldsymbol{\epsilon}_t, t) - \boldsymbol{\epsilon}_t\right\|^2\right]. \tag{5}$$

We use $\|\cdot\|_p$ to denote $\ell_p$-norm. Unless specified, the norm $\|\cdot\|$ refers to the $\ell_2$-norm $\|\cdot\|_2$. Since $\bar{\alpha}_t \to 0$ for $t \to T$, $\boldsymbol{x}_0$ is obtained by conducting the reverse diffusion process $p_{\boldsymbol{\theta}}(\boldsymbol{x}_t \mid \boldsymbol{x}_{t+1})$ starting from $\boldsymbol{x}_T \sim \mathcal{N}(0, \boldsymbol{I})$ and $\boldsymbol{\epsilon} \sim \mathcal{N}(0, \boldsymbol{I})$, under the learned model $\boldsymbol{\epsilon}_{\boldsymbol{\theta}}$ with

$$\boldsymbol{x}_t = \frac{1}{\sqrt{\alpha_{t+1}}}\left(\boldsymbol{x}_{t+1} - \frac{1 - \alpha_{t+1}}{\sqrt{1 - \bar{\alpha}_{t+1}}}\boldsymbol{\epsilon}_{\boldsymbol{\theta}}(\boldsymbol{x}_{t+1}, t+1)\right) + \sqrt{1 - \alpha_{t+1}}\boldsymbol{\epsilon}. \tag{6}$$

**Wasserstein Distance.** For integer $p > 0$, $\Gamma(\mu, \nu)$ as the set of union distributions with marginal $\mu$ and $\nu$, the Wasserstein $p$-distance (Villani et al., 2009) between distributions $\mu$ and $\nu$ with finite $p$-moments is

$$W_p^p(\mu, \nu) = \inf_{\gamma \in \Gamma(\mu, \nu)} \mathbb{E}_{(\boldsymbol{x}, \boldsymbol{y}) \sim \gamma} \|\boldsymbol{x} - \boldsymbol{y}\|_p^p. \tag{7}$$

## 4 ROBUSTNESS-DRIVEN ADVERSARIAL TRAINING OF DIFFUSION MODELS

In this section, we formally show that the success of DPM relies on specific conditions, i.e., $\boldsymbol{x}_t$ is close to $\boldsymbol{x}_{t+1}$. Next, to mitigate the drawbacks brought by the restriction, we propose to consider the distribution mismatch problem as discussed in Section 1, and connect the problem to a rewritten ELBO. Finally, we apply DRO for this ELBO to mitigate the distribution mismatch problem and finally link it to AT to be implemented in practice.

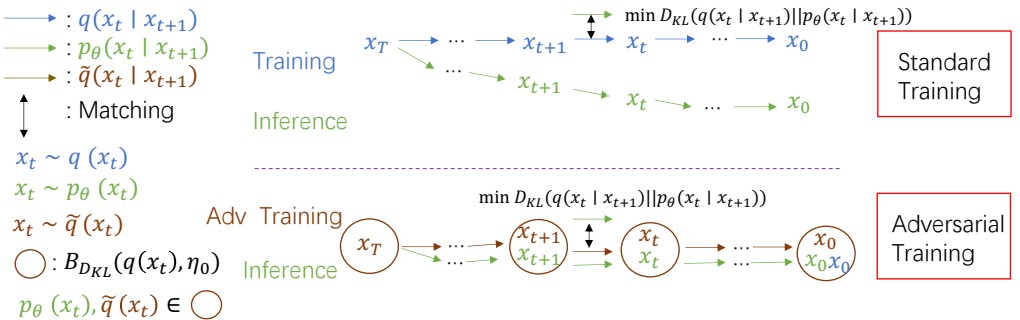

Figure 1: A comparison between standard training and the proposed distributional robust optimization in (12). When minimizing $D_{KL}(\tilde{q}_t(\boldsymbol{x}_t \mid \boldsymbol{x}_{t+1}) \parallel p_{\boldsymbol{\theta}}(\boldsymbol{x}_t \mid \boldsymbol{x}_{t+1}))$, the $\boldsymbol{x}_{t+1}$ is sampled from $\tilde{q}_t(\boldsymbol{x}_{t+1})$, such that both $\tilde{q}_t(\boldsymbol{x}_{t+1})$ in training stage and $p_{\boldsymbol{\theta}}(\boldsymbol{x}_{t+1})$ in inference stage are in $B_{D_{KL}}(q(x_{t+1}), \eta_0)$, so that $p_{\boldsymbol{\theta}}(\boldsymbol{x}_t)$ tends to locates in $B_{D_{KL}}(q(x_t), \eta_0)$ as well as $\tilde{q}_t(\boldsymbol{x}_t)$. Then, the distributional robustness captured by (12) guarantees the generated $p_{\boldsymbol{\theta}}(\boldsymbol{x}_t)$ always locates around $q(\boldsymbol{x}_t)$ for all $t$.

### 4.1 HOW DOES DPM WORKS IN PRACTICE?

Notably, minimizing (4) potentially obtains a sharp NLL under target distribution $q(\boldsymbol{x}_0)$. However, in the following proposition, we show that (4) also implicitly minimizes the NLL of each $\boldsymbol{x}_t$.

**Proposition 1.** *The minimization problem* (4) *is equivalent to minimizing an upper bound of* $\mathbb{E}_q[-\log p_{\boldsymbol{\theta}}(\boldsymbol{x}_t)]$ *for any* $0 \leq t \leq T$.

The proof is provided in Appendix A. It shows that though (4) is proposed to generate $\boldsymbol{x}_0 \sim q(\boldsymbol{x}_0)$, it also guides the model to generate $\boldsymbol{x}_t$ such that $p_{\boldsymbol{\theta}}(\boldsymbol{x}_t)$ approximates the ground-truth distribution $q(\boldsymbol{x}_t)$. The conclusion is nontrivial as minimizing the ELBO of NLL $\mathbb{E}_q[-\log p_{\boldsymbol{\theta}}(\boldsymbol{x}_0)]$ does not necessarily impose any restrictions on $\boldsymbol{x}_t$ for $t \geq 1$.

Next, we will further explain why (4) leads to a small NLL of $\boldsymbol{x}_t$. In $L_t$ of (4), $p_{\boldsymbol{\theta}}(\boldsymbol{x}_t \mid \boldsymbol{x}_{t+1})$ approximates $q(\boldsymbol{x}_t \mid \boldsymbol{x}_{t+1})$ with $\boldsymbol{x}_{t+1} \sim q(\boldsymbol{x}_{t+1})$ representing ground-truth data. Consequently, $p_{\boldsymbol{\theta}}(\boldsymbol{x}_t)$ approximates $q(\boldsymbol{x}_t)$ by recursively applying such a relationship as in the following proposition.

**Proposition 2.** *Suppose* $p_{\boldsymbol{\theta}}(\boldsymbol{x}_t \mid \boldsymbol{x}_{t+1})$ *matches* $q(\boldsymbol{x}_t \mid \boldsymbol{x}_{t+1})$ *well such that*

$$L_t = D_{KL}(q(\boldsymbol{x}_t \mid \boldsymbol{x}_{t+1}) \parallel p_{\boldsymbol{\theta}}(\boldsymbol{x}_t \mid \boldsymbol{x}_{t+1})) \leq \frac{\gamma}{T}, \tag{8}$$

*and the discrepancy satisfies* $D_{KL}(q(\boldsymbol{x}_T) \parallel p_{\boldsymbol{\theta}}(\boldsymbol{x}_T)) \leq \gamma_0$, *then for any* $0 \leq t \leq T$, *we have*

$$D_{KL}(q(\boldsymbol{x}_t) \parallel p_{\boldsymbol{\theta}}(\boldsymbol{x}_t)) \leq D_{KL}(q(\boldsymbol{x}_{t+1}) \parallel p_{\boldsymbol{\theta}}(\boldsymbol{x}_{t+1})) + L_t \leq \gamma_0 + \frac{(T-t)\gamma}{T}. \tag{9}$$

The results is similarly obtained in (Chen et al., 2023), while their result is applied for $D_{KL}(q(\boldsymbol{x}_0) \parallel p_{\boldsymbol{\theta}_0})$, which is narrowed compared with Proposition 2. The proof is provided in Appendix A, which formally explains why (4) results in $p_{\boldsymbol{\theta}}(\boldsymbol{x}_t)$ approximating $q(\boldsymbol{x}_t)$. However, this proposition is built upon small $L_t$, and notably, the error introduced by $L_t$ will be accumulated on the r.h.s. of (9), as it increases w.r.t. $t$. This phenomenon is caused by the *distribution mismatch problem* discussed in Section 1. Concretely, in (4), minimizing $L_t$ learns the transition probability $p_{\boldsymbol{\theta}}(\boldsymbol{x}_t \mid \boldsymbol{x}_{t+1})$ based on $\boldsymbol{x}_{t+1} \sim q(\boldsymbol{x}_{t+1})$, while in practice, $\boldsymbol{x}_t$ in (6) is generated from $\boldsymbol{x}_{t+1} \sim p_{\boldsymbol{\theta}}(\boldsymbol{x}_{t+1})$. The error between $p_{\boldsymbol{\theta}}(\boldsymbol{x}_{t+1})$ and $q(\boldsymbol{x}_{t+1})$ will propagates into the error between $p_{\boldsymbol{\theta}}(\boldsymbol{x}_t)$ and $q(\boldsymbol{x}_t)$ as in (9).

Therefore, owing to the existence of distribution mismatch, only if $L_t$ is minimized, the gap between $p_{\boldsymbol{\theta}}(\boldsymbol{x}_t)$ and $q(\boldsymbol{x}_t)$ can be guaranteed. However, the following proposition proved in Appendix A indicates that $L_t$ is theoretically minimized with restrictions.

**Proposition 3.** $L_t$ *in* (4) *is well minimized, only if* $q(\boldsymbol{x}_{t+1})$ *is Gaussian or* $\|\boldsymbol{x}_{t+1} - \boldsymbol{x}_t\| \to 0$.

In practice, the $q(\boldsymbol{x}_{t+1})$ is usually non-Gaussian. Besides, the gap $\|\boldsymbol{x}_{t+1} - \boldsymbol{x}_t\|$ is not necessarily small, especially for samplers with few sampling steps, e.g., DDIM (Song et al., 2022), DPM-Solver (Lu et al., 2022a). Therefore, in practice, the accumulated error in (9) caused by the distribution mismatch problem may become large, and degenerate the quality of $\boldsymbol{x}_0$.

## 4.2 DISTRIBUTIONAL ROBUSTNESS IN DPM

Inspired by the discussion above, we propose a new training objective as the sum of NLLs under $\boldsymbol{x}_t$,

$$\min_{\boldsymbol{\theta}} \mathcal{L}(\boldsymbol{\theta}) = \sum_{t=0}^{T} \mathbb{E}_q \left[ -\log p_{\boldsymbol{\theta}}(\boldsymbol{x}_t) \right]. \tag{10}$$

Then the following proposition constructs ELBOs for each of $\mathbb{E}_q[-\log p_{\boldsymbol{\theta}}(\boldsymbol{x}_t)]$.

**Proposition 4.** *For any distribution $\tilde{q}$ satisfies $\tilde{q}(\boldsymbol{x}_t) = q(\boldsymbol{x}_t)$ for specific $t$, we have*

$$\mathbb{E}_q \left[ -\log p_{\boldsymbol{\theta}}(\boldsymbol{x}_t) \right] \leq \underbrace{D_{KL}(\tilde{q}(\boldsymbol{x}_t \mid \boldsymbol{x}_{t+1}) \parallel p_{\boldsymbol{\theta}}(\boldsymbol{x}_t \mid \boldsymbol{x}_{t+1}))}_{L_t^{\tilde{q}}} + C, \tag{11}$$

*for a constant $C$ independent of $\boldsymbol{\theta}$.*

The proof is in Appendix A.2. This proposition generalizes the results in Proposition 1 since $\tilde{q}$ can be taken as $q$ in Proposition 1. During minimizing $L_t^{\tilde{q}}$, the transition probability $p_{\boldsymbol{\theta}}(\boldsymbol{x}_t \mid \boldsymbol{x}_{t+1})$ matches $\tilde{q}(\boldsymbol{x}_t \mid \boldsymbol{x}_{t+1})$, while $\boldsymbol{x}_{t+1} \sim \tilde{q}(\boldsymbol{x}_{t+1})$ in the training stage has no restriction. Thus, one may take $\tilde{q}(\boldsymbol{x}_{t+1}) \approx p_{\boldsymbol{\theta}}(\boldsymbol{x}_{t+1})$, then in $L_t^{\tilde{q}}$, $p_{\boldsymbol{\theta}}(\boldsymbol{x}_t \mid \boldsymbol{x}_{t+1})$ matches $\tilde{q}(\boldsymbol{x}_t \mid \boldsymbol{x}_{t+1})$ leads $p_{\boldsymbol{\theta}}(\boldsymbol{x}_t) \approx \tilde{q}(\boldsymbol{x}_t) = q(\boldsymbol{x}_t)$, which mitigates the distribution mismatch problem, when minimizing such $L_t^{\tilde{q}}$.

Unfortunately, for each $t$, obtaining such specific $\tilde{q}_t(\boldsymbol{x}_{t+1}) = p_{\boldsymbol{\theta}}(\boldsymbol{x}_{t+1})$ is computationally expensive (Li & van der Schaar, 2024), which prevents us using desired $\tilde{q}_t(\boldsymbol{x}_{t+1})$. However, we know $p_{\boldsymbol{\theta}}(\boldsymbol{x}_{t+1})$ is around $q(\boldsymbol{x}_{t+1})$. Therefore, by borrowing the idea from DRO (Shapiro, 2017), for each $t$, we propose to minimize the maximal value of $L_t^{\tilde{q}_t}$ over all possible $\tilde{q}_t(\boldsymbol{x}_{t+1})$ around $q(\boldsymbol{x}_{t+1})$. This leads to a small $L_t^{p_{\boldsymbol{\theta}}}$, as $p_{\boldsymbol{\theta}}(\boldsymbol{x}_{t+1})$ locates around $q(\boldsymbol{x}_{t+1})$, so that is included in the "maximal range". Technically, the DRO-based EBLO of (11) is formulated as follows. Here $p_{\boldsymbol{\theta}}(x_{t+1})$ is supposed in $B_{D_{KL}}(q(\boldsymbol{x}_{t+1}), \eta_0)$, and it captures the distributional robustness of $p_{\boldsymbol{\theta}}(\boldsymbol{x}_t \mid \boldsymbol{x}_{t+1})$ w.r.t. input $\boldsymbol{x}_{t+1}$.

$$\min_{\boldsymbol{\theta}} \sum_{t=0}^{T-1} L_t^{\mathrm{DRO}}(\boldsymbol{\theta}) = \min_{\boldsymbol{\theta}} \sum_{t=0}^{T-1} \sup_{\tilde{q}_t(\boldsymbol{x}_{t+1}) \in B_{D_{KL}}(q(\boldsymbol{x}_{t+1}), \eta_0)} D_{KL}(\tilde{q}_t(\boldsymbol{x}_t \mid \boldsymbol{x}_{t+1}) \parallel p_{\boldsymbol{\theta}}(\boldsymbol{x}_t \mid \boldsymbol{x}_{t+1})); \tag{12}$$
$$s.t. \qquad \tilde{q}_t(\boldsymbol{x}_t) = q(\boldsymbol{x}_t).$$

Here $\tilde{q}_t(\boldsymbol{x}_{t+1}) \in B_{D_{KL}}(q(\boldsymbol{x}_{t+1}), \eta_0)$ means $D_{KL}(q(\boldsymbol{x}_{t+1}) \parallel \tilde{q}_t(\boldsymbol{x}_{t+1})) \leq \eta_0$. By solving problem (12), if the desired $\tilde{q}_t(\boldsymbol{x}_{t+1}) = p_{\boldsymbol{\theta}}(\boldsymbol{x}_{t+1})$ is in $B_{D_{KL}}(q(\boldsymbol{x}_{t+1}), \eta_0)$, then the conditional probability in (12) transfers $\boldsymbol{x}_{t+1} \sim p_{\boldsymbol{\theta}}(\boldsymbol{x}_{t+1})$ to target $\boldsymbol{x}_t \sim q(\boldsymbol{x}_t)$ is learned, which mitigates the distribution mismatch problem. The theoretical clarification is in the following Proposition proved in Appendix A.2, which indicates that small DRO loss (12) guarantees the quality of generated $\boldsymbol{x}_0$.

**Proposition 5.** *If $L_t^{\mathrm{DRO}}(\boldsymbol{\theta}) \leq \eta_0$ in (12) for all $t$, and $D_{KL}(q(\boldsymbol{x}_T) \parallel p_{\boldsymbol{\theta}}(\boldsymbol{x}_T)) \leq \eta_0$, then $D_{KL}(q(x_0) \parallel p_{\boldsymbol{\theta}}(\boldsymbol{x}_0)) \leq \eta_0$.*

Up to now, we do not know how to compute the DRO-based training objective (12) we derived. Fortunately, the following theorem corresponds (12) to a "perturbed" noise prediction problem similar to (5). The theorem is proved in Appendix A.2.

**Theorem 1.** *There exists $\boldsymbol{\delta}_t$ depends on $\boldsymbol{x}_0$ and $\boldsymbol{\epsilon}_t$ makes (13) equivalent to problem (12).*

$$\min_{\boldsymbol{\theta}} \sum_{t=0}^{T-1} \mathbb{E}_{q(\boldsymbol{x}_0), \boldsymbol{\epsilon}_t} \left[ \left\| \boldsymbol{\epsilon}_{\boldsymbol{\theta}}(\sqrt{\bar{\alpha}_t} \boldsymbol{x}_0 + \sqrt{1 - \bar{\alpha}_t} \boldsymbol{\epsilon}_t + \boldsymbol{\delta}_t, t) - \boldsymbol{\epsilon}_t - \frac{\boldsymbol{\delta}_t}{\sqrt{1 - \bar{\alpha}_t}} \right\|^2 \right]. \tag{13}$$

This theorem connects the proposed DRO problem (12) with noise prediction problem (13). Naturally, we can solve (13), if we know the exact $\boldsymbol{\delta}_t$. Fortunately, we have the following proposition to characterize the range of $\boldsymbol{\delta}_t$, and it is proved in Appendix A.2.

**Proposition 6.** *For $\eta > 0$ and $\boldsymbol{\delta}_t$ in (13), $\|\boldsymbol{\delta}_t\|_1 \leq \eta$ holds with probability at least $1 - \sqrt{2(1 - \bar{\alpha}_t)/\eta}$.*

The proposition indicates that for any $\boldsymbol{\delta}_t$ depends on $\boldsymbol{x}_0, \boldsymbol{\epsilon}_t$ in (13), it is likely in a small range (measured under any $\ell_p$-norm, since they can bound each other in Euclidean space). Thus, to resolve (13) (so that (12)), we propose to directly consider the following adversarial training (Madry et al.,

2018) objective with the perturbation $\boldsymbol{\delta}$ is taken over its possible range as proved in Proposition 6, which captures the input (instead of distribution) robustness of model $\boldsymbol{\epsilon_\theta}$.

$$\min_{\boldsymbol{\theta}} \sum_{t=0}^{T-1} \mathbb{E}_{q(\boldsymbol{x}_0)} \left[ \mathbb{E}_{q(\boldsymbol{x}_t|\boldsymbol{x}_0)} \left[ \sup_{\boldsymbol{\delta}:\|\boldsymbol{\delta}\|\leq\eta} \left\| \boldsymbol{\epsilon_\theta}(\sqrt{\bar{\alpha}_t}\boldsymbol{x}_0 + \sqrt{1-\bar{\alpha}_t}\boldsymbol{\epsilon}_t + \boldsymbol{\delta}) - \boldsymbol{\epsilon}_t - \frac{\boldsymbol{\delta}}{\sqrt{1-\bar{\alpha}_t}} \right\|^2 \right] \right]. \quad (14)$$

We present a fine-grained connection between (14) and classical AT in Appendix C. Notably, our objective (14) is different from the ones in (Ning et al., 2023), whereas $\boldsymbol{\delta}$ in it is a Gaussian, and $\boldsymbol{\epsilon_\theta}$ predicts $\boldsymbol{\epsilon}_t$ instead of $\boldsymbol{\epsilon}_t + \boldsymbol{\delta}/\sqrt{1-\bar{\alpha}_t}$ as ours.

To make it clear, we summarize the rationale from DRO objective (12) to AT our objective (14). Since Theorem 1 shows solving (12) is equivalent to (13), which conducts noise prediction (5) with a perturbation $\boldsymbol{\delta}_t$ in a small range added (Proposition 6). Thus, we propose to minimize the maximal loss over the possible $\boldsymbol{\delta}_t$, which is indeed our AT objective (14).

## 5 ADVERSARIAL TRAINING UNDER CONSISTENCY MODEL

Although the DPM generates high-quality target data $\boldsymbol{x}_0$, the multi-step denoising process (6) requires numerous model evaluations, which can be computationally expensive. To resolve this, the diffusion-based consistency model (CM) is proposed in (Song et al., 2023). Consistency model $f_{\boldsymbol{\theta}}(\boldsymbol{x}_t, t)$ transfers $\boldsymbol{x}_t \sim q(\boldsymbol{x}_t)$ into a distribution that approximates the target $q(\boldsymbol{x}_0)$. $f_{\boldsymbol{\theta}}$ is optimized by the following consistency distillation (CD) loss [3]

$$\min_{\boldsymbol{\theta}} \mathcal{L}_{CD}(\boldsymbol{\theta}) = \sum_{t=0}^{T-1} \mathbb{E}_{\boldsymbol{x}_{t+1}\sim q(\boldsymbol{x}_{t+1})} \left[ d\left( f_{\boldsymbol{\theta}}(\Phi_t(\boldsymbol{x}_{t+1}), t), f_{\boldsymbol{\theta}}(\boldsymbol{x}_{t+1}, t+1) \right) \right], \quad (15)$$

where $\Phi_t(\boldsymbol{x}_{t+1})$ is a solution of a specific ordinary differential equation (ODE) ((37) in Appendix B) which is a deterministic function transfers $\boldsymbol{x}_{t+1}$ to $\boldsymbol{x}_t$, i.e., $\Phi_t(\boldsymbol{x}_{t+1}) \sim q(\boldsymbol{x}_t)$, and $d(\boldsymbol{x}, \boldsymbol{y})$ is a distance between $\boldsymbol{x}$ and $\boldsymbol{y}$ e.g., $\ell_1, \ell_2$ distance.

**Remark 1.** *In (Song et al., 2023; Luo et al., 2023), the noisy data $\boldsymbol{x}_t$ in (15) is described by an ODE (37) in Appendix B. However, we use the discrete $\boldsymbol{x}_t$ (1) here to unify the notations with Section 4. The two frameworks are mathematically equivalent as all $\boldsymbol{x}_t$ in (1) located in the trajectory of ODE in (Song et al., 2023). More details of this claim refer to Appendix B.*

Next, we use the following theorem to illustrate that solving problem (15) indeed creates $f_{\boldsymbol{\theta}}(\boldsymbol{x}_t, t)$ with distribution close target $q(\boldsymbol{x}_0)$. The theorem is proved in Appendix B.

**Theorem 2.** *For $\mathcal{L}_{CD}(\boldsymbol{\theta})$ in (15) with $d(\cdot, \cdot)$ is $\ell_2$ distance, then $\mathsf{W}_1(f_{\boldsymbol{\theta}}(\boldsymbol{x}_t, t), \boldsymbol{x}_0) \leq \sqrt{t\mathcal{L}_{CD}(\boldsymbol{\theta})}$ [4].*

Though solving problem (15) creates the desired CM $f_{\boldsymbol{\theta}}$, computing the exact $\Phi_t(\boldsymbol{x}_{t+1})$ involves solving an ODE as pointed out in Appendix B. Thus, in practice (Song et al., 2023; Luo et al., 2023), the $\Phi_t(\boldsymbol{x}_{t+1})$ is approximated by a computable numerical estimation $\hat{\Phi}_t(\boldsymbol{x}_{t+1}, \boldsymbol{\epsilon_\phi})$ of it, e.g., Euler ((42) in Appendix B.1) or DDIM (Song et al., 2023), where $\boldsymbol{\epsilon_\phi}$ is a pretrained noise prediction model as in (5). Therefore, the practical training objective of (15) becomes

$$\min_{\boldsymbol{\theta}} \sum_{t=0}^{T-1} \hat{\mathcal{L}}_{CD}(\boldsymbol{\theta}) = \mathbb{E}_{\boldsymbol{x}_{t+1}\sim q(\boldsymbol{z}_t)} \left[ d\left( f_{\boldsymbol{\theta}}(\hat{\Phi}_t(\boldsymbol{x}_{t+1}, \boldsymbol{\epsilon_\phi}), t), f_{\boldsymbol{\theta}}(\boldsymbol{x}_{t+1}, t+1) \right) \right]. \quad (16)$$

In (16), $\hat{\Phi}_t(\boldsymbol{x}_{t+1}, \boldsymbol{\epsilon_\phi})$ is an estimation to $\Phi_t(\boldsymbol{x}_{t+1})$, which causes an inaccurate training objective $\hat{\mathcal{L}}_{CD}$ in (16), compared with target $\mathcal{L}_{CD}$ (15). Thus, this results in the distribution mismatch problem in CM, as in DPM of Section 4. However, similar to Section 4.2, if we train $f_{\boldsymbol{\theta}}$ with robustness to the gap between $\hat{\Phi}_t(\boldsymbol{x}_{t+1}, \boldsymbol{\epsilon_\phi})$ and $\Phi_t(\boldsymbol{x}_{t+1})$, the distribution mismatch problem in CM is mitigated.

Technically, suppose $\Phi_t(\boldsymbol{x}_{t+1}) = \hat{\Phi}_t(\boldsymbol{x}_{t+1}, \boldsymbol{\epsilon_\phi}) + \boldsymbol{\delta}_t(\boldsymbol{x}_{t+1})$, we can consider minimizing the following adversarial training objective of CM, if $\|\boldsymbol{\delta}_t(\boldsymbol{x}_{t+1})\| \leq \eta$ uniformly over $t$, for some constant $\eta$,

---

[3] In practice, (15) is updated under target model $f_{\boldsymbol{\theta}^-}(\Phi_t(\boldsymbol{x}_{t+1}), t)$ with exponential moving average (EMA) $\boldsymbol{\theta}^-$ under a stop gradient operation. (Song et al., 2023) find that it greatly stabilizes the training process. In this section, we focus on the theory of consistency model and still use $\boldsymbol{\theta}$ in formulas.

[4] Here $\mathsf{W}_1(f_{\boldsymbol{\theta}}(\boldsymbol{x}_t, t), \boldsymbol{x}_0)$ is the Wasserstein 1-distance between distributions of $f_{\boldsymbol{\theta}}(\boldsymbol{x}_t, t)$ and $\boldsymbol{x}_0$.

---

**Algorithm 1** Adversarial Training for Diffusion Model

---

1: **Input:** dataset $\mathcal{D}$, model parameter $\boldsymbol{\theta}$, learning rate $\kappa$, loss weighting $\lambda(\cdot)$, adversarial steps $K$, adversarial learning rate $\alpha$
2: **while** do not converge **do**
3:     Sample $\boldsymbol{x} \sim \mathcal{D}$ and $t \sim \mathcal{U}[1, T]$
4:     Sample $\boldsymbol{\epsilon} \sim \mathcal{N}(\boldsymbol{0}, \boldsymbol{I})$
5:     $\boldsymbol{\delta} \leftarrow \boldsymbol{0}$
6:     **for** $i = 1, 2, \ldots, K$ **do**
7:         $\mathcal{L} \leftarrow \left\| \boldsymbol{\epsilon_\theta}(\sqrt{\bar{\alpha}_t}\boldsymbol{x}_0 + \sqrt{1 - \bar{\alpha}_t}\boldsymbol{\epsilon} + \boldsymbol{\delta}) - \boldsymbol{\epsilon} - \frac{\boldsymbol{\delta}}{\sqrt{1-\bar{\alpha}_t}} \right\|^2$ in (14)
8:         $\boldsymbol{\delta} \leftarrow \boldsymbol{\delta} + \alpha \cdot \frac{\nabla_{\boldsymbol{\delta}}\mathcal{L}}{\|\nabla_{\boldsymbol{\delta}}\mathcal{L}\|}$                 ▷ *maximize perturbation*
9:         $\boldsymbol{\theta} \leftarrow \boldsymbol{\theta} - \kappa \cdot \nabla_{\boldsymbol{\theta}}\mathcal{L}$                      ▷ *update model*
10:     **end for**
11: **end while**

---

**Algorithm 2** Adversarial Training for Consistency Distillation

---

1: **Input:** dataset $\mathcal{D}$, initial model parameter $\boldsymbol{\theta}$, learning rate $\kappa$, pretrained noise prediction model $\boldsymbol{\epsilon_\phi}$, ODE solver $\hat{\Phi}.(\cdot, \boldsymbol{\epsilon_\phi})$, metric $d(\cdot, \cdot)$, loss weighting $\lambda(\cdot)$, target model EMA $\mu$, adversarial steps $K$, adversarial learning rate $\alpha$
2: $\boldsymbol{\theta}^- \leftarrow \boldsymbol{\theta}$
3: **while** do not converge **do**
4:     Sample $\boldsymbol{x} \sim \mathcal{D}$ and $t \sim \mathcal{U}[0, T-1]$
5:     Sample $\boldsymbol{x}_{t+1}$ from (1)
6:     $\boldsymbol{\delta} \leftarrow \boldsymbol{0}$
7:     **for** $i = 1, 2, \ldots, K$ **do**
8:         $\mathcal{L} \leftarrow \lambda(t)d(f_{\boldsymbol{\theta}}(\boldsymbol{x}_{t+1}, t+1), f_{\boldsymbol{\theta}^-}(\hat{\Phi}_t(\boldsymbol{x}_{t+1}, \boldsymbol{\epsilon_\phi}) + \boldsymbol{\delta}, t))$ in (17)
9:         $\boldsymbol{\delta} \leftarrow \boldsymbol{\delta} + \alpha \cdot \frac{\nabla_{\boldsymbol{\delta}}\mathcal{L}}{\|\nabla_{\boldsymbol{\delta}}\mathcal{L}\|}$            ▷ *maximize perturbation*
10:         $\boldsymbol{\theta} \leftarrow \boldsymbol{\theta} - \kappa \cdot \nabla_{\boldsymbol{\theta}}\mathcal{L}$                ▷ *update model*
11:         $\boldsymbol{\theta}^- \leftarrow \text{stopgrad}(\mu\boldsymbol{\theta}^- + (1-\mu)\boldsymbol{\theta})$
12:     **end for**
13: **end while**

---

so that the target $\Phi_t(\boldsymbol{x}_{t+1})$ is included in the maximal range as well.

$$\hat{\mathcal{L}}_{CD}^{Adv}(\boldsymbol{\theta}) = \sum_{t=0}^{T-1} \mathbb{E}_{\boldsymbol{x}_{t+1}} \left[ \sup_{\|\boldsymbol{\delta}\| \leq \eta} d\left( f_{\boldsymbol{\theta}}(\hat{\Phi}_t(\boldsymbol{x}_{t+1}, \boldsymbol{\epsilon_\phi}) + \boldsymbol{\delta}, t), f_{\boldsymbol{\theta}}(\boldsymbol{x}_{t+1}, t+1) \right) \right]. \tag{17}$$

By doing so, the learned model $f_{\boldsymbol{\theta}}$ can be robust to the perturbation brought by $\boldsymbol{\delta}_t(\boldsymbol{x}_{t+1})$, so that results in a small $\mathcal{L}_{CD}(\boldsymbol{\theta})$, as well as the small $\mathsf{W}_1(f_{\boldsymbol{\theta}}(\boldsymbol{x}_T, T), \boldsymbol{x}_0)$ as proved in Theorem 2. Next, we use the following theorem to show that $\|\boldsymbol{\delta}_t(\boldsymbol{x}_{t+1})\|$ is indeed small, and minimizing $\hat{\mathcal{L}}_{CD}^{Adv}(\boldsymbol{\theta})$ results in $f_{\boldsymbol{\theta}}(\boldsymbol{x}_T, T)$ with distribution approximates $\boldsymbol{x}_0$.

**Theorem 3.** *Under proper regularity conditions, for $0 \leq t < T$, we have $\mathbb{E}_{\boldsymbol{x}_{t+1}}[\|\boldsymbol{\delta}_t(\boldsymbol{x}_{t+1})\|] \leq o(1)$. On the other hand, it holds*

$$\mathsf{W}_1(f_{\boldsymbol{\theta}}(\boldsymbol{x}_T, T), \boldsymbol{x}_0) \leq \sqrt{T\hat{\mathcal{L}}_{CD}^{Adv}(\boldsymbol{\theta})} + o(1). \tag{18}$$

The theorem is proved in Appendix B.1, and it indicates that using the proposed adversarial training objective (17) of CM indeed guarantees the learned CM transfers $\boldsymbol{x}_T$ into data from $q(\boldsymbol{x}_0)$.

## 6 EXPERIMENTS

### 6.1 ALGORITHMS

In the standard adversarial training method like Projected Gradient Descent (PGD) (Madry et al., 2018), the perturbation $\boldsymbol{\delta}$ is constructed by implementing numbers (3-8) of gradient ascents to $\boldsymbol{\delta}$

before updating the model, which slows down the training process. To resolve this, we adopt an efficient implementation (Shafahi et al., 2019) in Algorithms 1, 2 to solve AT (14) and (17) of DPM and CM, *which has similar computational cost compared to standard training*, and significantly accelerate standard AT. Notably, unlike PGD, in Algorithms 1 and 2, every maximization step of perturbation $\delta$ follows an update step of the model $\theta$. Thus, the efficient AT do not require further back propagations to construct adversarial samples as in PGD. We provide a comparison between our efficient AT and standard AT (PGD) with the same update iterations of model $\theta$ in Appendix G.1. Moreover, we observe that efficient AT can yield comparable and even better performance than PGD while accelerating the training ($2.6\times$ speed-up), further verifying the benefits of our efficient AT. [5]

## 6.2 Performance on DPM

**Settings.** The experiments are conducted on the unconditional generation on `CIFAR-10` $32\times32$ (Krizhevsky & Hinton, 2009) and the class-conditional generation on `ImageNet` $64 \times 64$ (Deng et al., 2009). Our model and training pipelines in adopted from ADM (Dhariwal & Nichol, 2021) paper, where ADM is a UNet-type network (Ronneberger et al., 2015), with strong performance in image generation under diffusion model.

To save training costs, our methods and baselines are fine-tuned from pretrained models, rather than training from scratch. By doing so, we can efficiently assess the performance of methods, which is more practical for general scenarios. We also explore training from scratch in Appendix G.2, which also verifies the effectiveness of our method in this regime. During training, we fine-tune the pretrained models (details are in Appendix E.1) with batch size 128 for 150K iterations under learning rate 1e-4 on `CIFAR-10`, and batch size 1024 for 50K iterations under learning rate of 3e-4 on `ImageNet`. For the hyperparameters of AT, we select the adversarial learning rate $\alpha$ from $\{0.05, 0.1, 0.5\}$ and the adversarial step $K$ from $\{3, 5\}$. More details are in Appendix E.1.

We use the Frechet Inception Distance (FID) (Heusel et al., 2017) to evaluate image quality. Unless otherwise specified, 50K images are sampled for evaluation. Other results of metric Classification Accuracy Score (CAS) (Ravuri & Vinyals, 2019), sFID, Inception Score, Precision, and Recall are in Appendix F.1 and F.4 for comprehensive evaluation.

**Baselines.** For experiments on diffusion models, we consider the following baselines. 1): the original pretrained model. Compared with it, we verify whether the models are overfitting during fine-tuning. 2): continue fine-tuning the pretrained model, which is fine-tuned with the standard diffusion objective (5). Compared to it, we validate whether performance improvements come only from more training costs. We also compare with the existing typical method to alleviate the DPM distribution mismatch, 3): ADM-IP (Ning et al., 2023), which adds a Gaussian perturbation to the input data to simulate mismatch errors during the training process. The last two fine-tuning baselines are based on **the same** pretrained model and hyperparameters as in the original literature.

**Results.** To verify the effectiveness of our AT method, we conduct experiments with four diffusion samplers: IDDPM (Dhariwal & Nichol, 2021), DDIM (Song et al., 2022), DPM-Solver (Lu et al., 2022b), and ES (Ning et al., 2024) under various NFEs. The sampler choices contain the three most popular samplers: IDDPM, DDIM, DPM-Solver, and ES, a sampler that scales down the norm of predicted noise to mitigate the distribution mismatch from the perspective of sampling. The experimental results of `CIFAR-10` and `ImageNet` are shown in Table 1 and Table 2, respectively. Results of more than hundreds of NFEs are shown in Appendix F.3

As can be seen, the proposed AT for DPM significantly improves the performance of the original pretrained model and outperforms the other baselines (continue fine-tuning and ADM-IP) overall for all diffusion samplers and NFEs we take. Moreover, we have the following observarions.

1): Fewer (practically used) sampling steps (5,10) will result in larger mismatching errors, while our AT method demonstrates significant improvements in this regime across various samplers, e.g., AT improves FID 27.72 to 17.36 under 5 NFEs DPM-Solver on `ImageNet`. This suggests that our method is indeed effective in alleviating the distribution mismatch of DPM. The results also indicate that our method consistently beats the baseline methods, regardless of stochastic (IDDPM)

---

[5]For the experts in AT, they would recognize that the AT in Algorithms 1, 2 actually constructs the adversarial augmented data to improve the performance of the model (Zhu et al., 2020; Jiang et al., 2020; Yi et al., 2021).

Table 1: Sample quality measured by FID ↓ of different sampling methods of DPM under different NFEs on `CIFAR10` 32x32. All models are trained with same iterations (computational costs).

(a) IDDPM

| Methods \ NFEs | 5 | 8 | 10 | 20 | 50 |
|---|---|---|---|---|---|
| ADM (original) | 37.99 | 26.75 | 22.62 | 10.52 | 4.55 |
| ADM (finetune) | **36.91** | 26.06 | 21.94 | 10.58 | 4.34 |
| ADM-IP | 47.57 | 26.91 | 20.09 | 7.81 | 3.42 |
| ADM-AT (Ours) | 37.15 | **23.59** | **15.88** | **6.60** | **3.34** |

(b) DDIM

| Methods \ NFEs | 5 | 8 | 10 | 20 | 50 |
|---|---|---|---|---|---|
| ADM (original) | 34.28 | 14.34 | 11.66 | 7.00 | 4.68 |
| ADM (finetune) | 29.30 | 15.08 | 12.06 | 6.80 | 4.15 |
| ADM-IP | 43.15 | 15.72 | 10.47 | 4.58 | 4.89 |
| ADM-AT (Ours) | **26.38** | **12.98** | **9.30** | **4.40** | **3.07** |

(c) ES

| Methods \ NFEs | 5 | 8 | 10 | 20 | 50 |
|---|---|---|---|---|---|
| ADM (original) | 82.18 | 29.28 | 17.73 | 5.11 | 2.70 |
| ADM (finetune) | 63.46 | 24.80 | 17.03 | 5.19 | 2.52 |
| ADM-IP | 91.10 | 31.44 | 18.72 | 5.19 | 2.89 |
| ADM-AT (Ours) | **41.07** | **21.62** | **14.68** | **4.36** | **2.48** |

(d) DPM-Solver

| Methods \ NFEs | 5 | 8 | 10 | 20 | 50 |
|---|---|---|---|---|---|
| ADM (original) | 23.95 | 8.00 | 5.46 | 3.46 | 3.14 |
| ADM (finetune) | 22.98 | 7.61 | 5.29 | 3.41 | 3.12 |
| ADM-IP | 43.83 | 6.70 | 6.80 | 9.78 | 10.91 |
| ADM-AT (Ours) | **18.40** | **5.84** | **4.81** | **3.28** | **3.01** |

Table 2: Sample quality measured by FID ↓ of different sampling methods of DPM under different NFEs on `ImageNet` 64x64. All models are trained with the same iterations (computational costs).

(a) IDDPM

| Methods \ NFEs | 5 | 8 | 10 | 20 | 50 |
|---|---|---|---|---|---|
| ADM (original) | 76.92 | 33.74 | 27.63 | 12.85 | 5.30 |
| ADM (finetune) | 78.87 | 33.99 | 27.82 | 12.80 | 5.26 |
| ADM-IP | 67.12 | 29.96 | 22.60 | 8.66 | **3.83** |
| ADM-AT (Ours) | **45.65** | **23.79** | **19.18** | **8.28** | 4.01 |

(b) DDIM

| Methods \ NFEs | 5 | 8 | 10 | 20 | 50 |
|---|---|---|---|---|---|
| ADM (original) | 60.07 | 20.10 | 14.97 | 8.41 | 5.65 |
| ADM (finetune) | 60.32 | 20.26 | 15.04 | 8.32 | 5.48 |
| ADM-IP | 76.51 | 26.25 | 18.05 | 8.40 | 6.94 |
| ADM-AT (Ours) | **43.04** | **16.08** | **12.15** | **6.20** | **4.67** |

(c) ES

| Methods \ NFEs | 5 | 8 | 10 | 20 | 50 |
|---|---|---|---|---|---|
| ADM (original) | 71.31 | 28.97 | 21.10 | 8.23 | 3.76 |
| ADM (finetune) | 72.30 | 29.24 | 21.58 | 8.25 | 3.64 |
| ADM-IP | 88.37 | 33.91 | 23.32 | 7.80 | 3.54 |
| ADM-AT (Ours) | **43.95** | **19.57** | **14.12** | **6.16** | **3.45** |

(d) DPM-Solver

| Methods \ NFEs | 5 | 8 | 10 | 20 | 50 |
|---|---|---|---|---|---|
| ADM (original) | 27.72 | 10.06 | 7.21 | 4.69 | 4.24 |
| ADM (finetune) | 27.82 | 9.97 | 7.22 | 4.64 | **4.15** |
| ADM-IP | 32.43 | 9.94 | 8.87 | 9.16 | 9.68 |
| ADM-AT (Ours) | **17.36** | **6.55** | **5.78** | **4.56** | 4.34 |

or deterministic samplers (DDIM, DPM-Solver). 2): The ES sampler results show that our AT is orthogonal to the sampling-based method to mitigate the distribution mismatch problem and can be combined to further alleviate the issue. Notably, we further verify in Appendix G.2 that our methods will not slow the convergence unlike AT in classification (Madry et al., 2018). We also perform ablation analysis of hyperparameters in our AT framework in Appendix G.3.

## 6.3 PERFORMANCE ON LATENT CONSISTENCY MODELS

**Settings.** We further evaluate the proposed AT for consistency models on text-to-image generation tasks with Latent Consistency Models (Luo et al., 2023) Stable Diffusion (SD) v1.5 (Rombach et al., 2022) backbone, which generates $512 \times 512$ images. Both our AT and the original LCM training (baseline) are trained from scratch with the same hyperparameters (the IP method (Ning et al., 2023) is not applied straightforwardly). The training set is LAION-Aesthetics-6.5+ (Schuhmann et al., 2022) with hyperparameters following Song et al. (2023); Luo et al. (2023). We select the adversarial learning rate $\alpha$ from $\{0.02, 0.05\}$ and adversarial step $K$ from $\{2, 3\}$. The models are trained with a batch size of 64 for 100K iterations. More details are shown in Appendix E.2.

Following Luo et al. (2023) and Chen et al. (2024), we evaluate models on MS-COCO 2014 (Lin et al., 2014a) at a resolution of $512 \times 512$ by randomly drawing 30K prompts from its validation set. Then, we report the FID between the generated samples under these prompts and the reference samples from the full validation set following Saharia et al. (2022). We also report CLIP scores (Hessel et al., 2021) to evaluate the text-image alignment by CLIP-ViT-B/16.

Table 3: Results of LCM on MS-COCO 2014 validation set at 512×512 resolution in terms of FID ↓ and CLIP score ↑. All models are trained with the same setting (computational costs).

| Methods | FID ↓ | | | | CLIP Score ↑ | | | |
|---|---|---|---|---|---|---|---|---|
| | 1 step | 2 step | 4 step | 8 step | 1 step | 2 step | 4 step | 8 step |
| LCM | 25.43 | 12.61 | 11.61 | 12.62 | 29.25 | 30.24 | 30.40 | 30.47 |
| LCM-AT (Ours) | **23.34** | **11.28** | **10.31** | **10.68** | **29.63** | **30.43** | **30.49** | **30.53** |

**Results.** The methods are evaluated under various sampling steps in Table 3, which shows that the LCM with AT consistently improves FID under various sampling steps. Besides, though the AT is not specified to improve text-image alignment, we observe that it has comparable or even better CLIP scores across various sampling steps, which shows that AT will not degenerate text-image alignment.

## 7 CONCLUSION

In this paper, we novelly introduce efficient Adversarial Training (AT) in the training of DPM and CM to mitigate the issue of distribution mismatch between training and sampling. We conduct an in-depth analysis of the DPM training objective and systematically characterize the distribution mismatch problem. Furthermore, we prove that the training objective of CM similarly faces the distribution mismatch issue. We theoretically prove that DRO can mitigate the mismatch for both DPM and CM, which is equivalent to conducting AT. Experiments on image generation and text-to-image generation benchmarks verify the effectiveness of the proposed AT method in alleviating the distribution mismatch of DPM and CM.

ACKNOWLEDGMENTS

We thank anonymous reviewers for insightful feedback that helped improve the paper. Zekun Wang, Ming Liu, Bing Qin are supported by the National Science Foundation of China (U22B2059, 62276083), the Human-Machine Integrated Consultation System for Cardiovascular Diseases (2023A003). They also appreciate the support from China Mobile Group Heilongjiang Co., Ltd.

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

# A PROOFS IN SECTION 4

In this section, we present the proofs of the results in Section 4.

## A.1 PROOFS IN SECTION 4.2

**Proposition 1.** *The minimization problem* (4) *is equivalent to minimizing an upper bound of* $\mathbb{E}_q[-\log p_{\boldsymbol{\theta}}(\boldsymbol{x}_t)]$ *for any* $0 \leq t \leq T$.

*Proof.* We prove the first equivalence, by Jensen's inequality. For any $0 \leq t < T$, we have

$$
\begin{aligned}
&- \mathbb{E}_q\left[\log p_{\boldsymbol{\theta}}(\boldsymbol{x}_t)\right] \\
\leq & \mathbb{E}_q\left[-\log \frac{p_{\boldsymbol{\theta}}(\boldsymbol{x}_{t:T})}{q(\boldsymbol{x}_{t+1:T} \mid \boldsymbol{x}_t)}\right] \\
= & \mathbb{E}_q\left[-\log p_{\boldsymbol{\theta}}(\boldsymbol{x}_T) - \sum_{t \leq s < T} \log \frac{p_{\boldsymbol{\theta}}(\boldsymbol{x}_s \mid \boldsymbol{x}_{s+1})}{q(\boldsymbol{x}_{s+1} \mid \boldsymbol{x}_s)}\right] \\
= & \mathbb{E}_q\left[-\log p_{\boldsymbol{\theta}}(\boldsymbol{x}_T) - \sum_{t \leq s < T} \log \frac{p_{\boldsymbol{\theta}}(\boldsymbol{x}_s \mid \boldsymbol{x}_{s+1})}{q(\boldsymbol{x}_s \mid \boldsymbol{x}_{s+1})} \cdot \frac{q(\boldsymbol{x}_s)}{q(\boldsymbol{x}_{s+1})}\right] \\
= & \mathbb{E}_q\left[-\log \frac{p_{\boldsymbol{\theta}}(\boldsymbol{x}_T)}{q(\boldsymbol{x}_T)} - \sum_{t \leq s < T} \log \frac{p_{\boldsymbol{\theta}}(\boldsymbol{x}_s \mid \boldsymbol{x}_{s+1})}{q(\boldsymbol{x}_s \mid \boldsymbol{x}_{s+1})} - \log q(\boldsymbol{x}_t)\right] \\
= & D_{KL}(q(\boldsymbol{x}_T) \| p_{\boldsymbol{\theta}}(\boldsymbol{x}_T)) + \mathbb{E}_q\left[\sum_{s=t}^{T-1} \underbrace{D_{KL}(q(\boldsymbol{x}_s \mid \boldsymbol{x}_{s+1}) \| p_{\boldsymbol{\theta}}(\boldsymbol{x}_s \mid \boldsymbol{x}_{s+1}))}_{L_t}\right] + H(\boldsymbol{x}_t)
\end{aligned}
\tag{19}
$$

Taking $t = 0$, we prove the first equivalence. Besides that, the entropy $H(x_t)$ of $\boldsymbol{x}_t$ is a constant for $\boldsymbol{\theta}$ given data distribution $\boldsymbol{x}_0$ for any $0 \leq t < T$. The second conclusion holds due to the non-negative property of KL-divergence. □

**Proposition 2.** *Suppose* $p_{\boldsymbol{\theta}}(\boldsymbol{x}_t \mid \boldsymbol{x}_{t+1})$ *matches* $q(\boldsymbol{x}_t \mid \boldsymbol{x}_{t+1})$ *well such that*

$$
L_t = D_{KL}(q(\boldsymbol{x}_t \mid \boldsymbol{x}_{t+1}) \| p_{\boldsymbol{\theta}}(\boldsymbol{x}_t \mid \boldsymbol{x}_{t+1})) \leq \frac{\gamma}{T},
\tag{8}
$$

*and the discrepancy satisfies* $D_{KL}(q(\boldsymbol{x}_T) \| p_{\boldsymbol{\theta}}(\boldsymbol{x}_T)) \leq \gamma_0$, *then for any* $0 \leq t \leq T$, *we have*

$$
D_{KL}(q(\boldsymbol{x}_t) \| p_{\boldsymbol{\theta}}(\boldsymbol{x}_t)) \leq D_{KL}(q(\boldsymbol{x}_{t+1}) \| p_{\boldsymbol{\theta}}(\boldsymbol{x}_{t+1})) + L_t \leq \gamma_0 + \frac{(T-t)\gamma}{T}.
\tag{9}
$$

*Proof.* We have the following decomposition due to the chain rule of KL-divergence

$$
\begin{aligned}
D_{KL}(q(\boldsymbol{x}_t, \boldsymbol{x}_{t+1}) \| p_{\boldsymbol{\theta}}(\boldsymbol{x}_t, \boldsymbol{x}_{t+1})) &= D_{KL}(q(\boldsymbol{x}_t \mid \boldsymbol{x}_{t+1}) \| p_{\boldsymbol{\theta}}(\boldsymbol{x}_t \mid \boldsymbol{x}_{t+1})) + D_{KL}(q(\boldsymbol{x}_{t+1}) \| p_{\boldsymbol{\theta}}(\boldsymbol{x}_{t+1})) \\
&= D_{KL}(q(\boldsymbol{x}_{t+1} \mid \boldsymbol{x}_t) \| p_{\boldsymbol{\theta}}(\boldsymbol{x}_{t+1} \mid \boldsymbol{x}_t)) + D_{KL}(q(\boldsymbol{x}_t) \| p_{\boldsymbol{\theta}}(\boldsymbol{x}_t)),
\end{aligned}
\tag{20}
$$

The transition probability $p_{\boldsymbol{\theta}}(\boldsymbol{x}_t \mid \boldsymbol{x}_{t+1})$ matches $q(\boldsymbol{x}_t \mid \boldsymbol{x}_{t+1})$, so that the above equality implies

$$
\begin{aligned}
&D_{KL}(q(\boldsymbol{x}_t) \| p_{\boldsymbol{\theta}}(\boldsymbol{x}_t)) \\
= & D_{KL}(q(\boldsymbol{x}_{t+1}) \| p_{\boldsymbol{\theta}}(\boldsymbol{x}_{t+1})) + D_{KL}(q(\boldsymbol{x}_t \mid \boldsymbol{x}_{t+1}) \| p_{\boldsymbol{\theta}}(\boldsymbol{x}_t \mid \boldsymbol{x}_{t+1})) - D_{KL}(q(\boldsymbol{x}_{t+1} \mid \boldsymbol{x}_t) \| p_{\boldsymbol{\theta}}(\boldsymbol{x}_{t+1} \mid \boldsymbol{x}_t)) \\
\leq & D_{KL}(q(\boldsymbol{x}_{t+1}) \| p_{\boldsymbol{\theta}}(\boldsymbol{x}_{t+1})) + \frac{\gamma}{T}.
\end{aligned}
\tag{21}
$$

The proposition holds due to initial condition $D_{KL}(q(\boldsymbol{x}_T) \| p_{\boldsymbol{\theta}}(\boldsymbol{x}_T)) \leq \gamma_0$ and simple induction. □

**Proposition 3.** *$L_t$ in* (4) *is well minimized, only if* $q(\boldsymbol{x}_{t+1})$ *is Gaussian or* $\|\boldsymbol{x}_{t+1} - \boldsymbol{x}_t\| \rightarrow 0$.

*Proof.* Due to Bayes' rule, we have

$$q(\boldsymbol{x}_t \mid \boldsymbol{x}_{t+1}) = \frac{q(\boldsymbol{x}_{t+1} \mid \boldsymbol{x}_t)q(\boldsymbol{x}_t)}{q(\boldsymbol{x}_{t+1})}$$

$$\propto \exp\left(-\frac{\left\|\boldsymbol{x}_{t+1} - \sqrt{\alpha_{t+1}}\boldsymbol{x}_t\right\|^2}{2(1-\alpha_{t+1})} + \log q(\boldsymbol{x}_t) - \log q(\boldsymbol{x}_{t+1})\right)$$

$$\propto \exp\left(-\frac{\left\|\boldsymbol{x}_{t+1} - \sqrt{\alpha_{t+1}}\boldsymbol{x}_t\right\|^2}{2(1-\alpha_{t+1})} + \langle\nabla_{\boldsymbol{x}}\log q(\boldsymbol{x}_{t+1}), \boldsymbol{x}_t - \boldsymbol{x}_{t+1}\rangle\right) \cdot$$

$$\exp\left(\frac{1}{2}(\boldsymbol{x}_t - \boldsymbol{x}_{t+1})^\top \nabla_{\boldsymbol{x}}^2 \log q(\boldsymbol{x}_{t+1})(\boldsymbol{x}_t - \boldsymbol{x}_{t+1}) + O(\|\boldsymbol{x}_{t+1} - \boldsymbol{x}_t\|^3)\right).$$

(22)

As can be seen, the conditional probability can be approximated by Gaussian only if $\nabla_{\boldsymbol{x}}^3 \log q(\boldsymbol{x}_{t+1})$ is zero or $\|\boldsymbol{x}_{t+1} - \boldsymbol{x}_t\|^3$ is extremely small with high probability. The two conditions can be respectively satisfied when $q(\boldsymbol{x}_t)$ is a Gaussian or $\boldsymbol{x}_t$ close to $\boldsymbol{x}_{t+1}$. $\qquad\square$

## A.2 Proofs in Section 4.2

**Proposition 4.** *For any distribution $\tilde{q}$ satisfies $\tilde{q}(\boldsymbol{x}_t) = q(\boldsymbol{x}_t)$ for specific t, we have*

$$\mathbb{E}_q\left[-\log p_{\boldsymbol{\theta}}(\boldsymbol{x}_t)\right] \leq \underbrace{D_{KL}(\tilde{q}(\boldsymbol{x}_t \mid \boldsymbol{x}_{t+1}) \parallel p_{\boldsymbol{\theta}}(\boldsymbol{x}_t \mid \boldsymbol{x}_{t+1}))}_{L_t^{\tilde{q}}} + C, \tag{11}$$

*for a constant $C$ independent of $\boldsymbol{\theta}$.*

*Proof.* W.o.l.g., suppose $p_{\boldsymbol{\theta}}(\boldsymbol{x}_t, \boldsymbol{x}_{t+1}) = p_{\boldsymbol{\theta}}(\boldsymbol{x}_t \mid \boldsymbol{x}_{t+1})q(\boldsymbol{x}_{t+1})$ and $\tilde{q}(\boldsymbol{x}_t, \boldsymbol{x}_{t+1}) = \tilde{q}(\boldsymbol{x}_{t+1} \mid \boldsymbol{x}_t)q(\boldsymbol{x}_t)$. By Jensen's inequality, we have

$$\mathbb{E}_q\left[-\log p_{\boldsymbol{\theta}}(\boldsymbol{x}_t)\right]$$

$$= -\int q(\boldsymbol{x}_t)\left(\log\int p_{\boldsymbol{\theta}}(\boldsymbol{x}_t, \boldsymbol{x}_{t+1})d\boldsymbol{x}_{t+1}\right)d\boldsymbol{x}_t$$

$$= -\int q(\boldsymbol{x}_t)\left(\log\int \frac{p_{\boldsymbol{\theta}}(\boldsymbol{x}_t, \boldsymbol{x}_{t+1})}{\tilde{q}(\boldsymbol{x}_{t+1} \mid \boldsymbol{x}_t)}\tilde{q}(\boldsymbol{x}_{t+1} \mid \boldsymbol{x}_t)d\boldsymbol{x}_{t+1}\right)d\boldsymbol{x}_t$$

$$\leq -\int q(\boldsymbol{x}_t)\left(\int\log \frac{p_{\boldsymbol{\theta}}(\boldsymbol{x}_t, \boldsymbol{x}_{t+1})}{\tilde{q}(\boldsymbol{x}_{t+1} \mid \boldsymbol{x}_t)}\tilde{q}(\boldsymbol{x}_{t+1} \mid \boldsymbol{x}_t)d\boldsymbol{x}_{t+1}\right)d\boldsymbol{x}_t$$

$$= -\int q(\boldsymbol{x}_t)\left(\int\tilde{q}(\boldsymbol{x}_{t+1} \mid \boldsymbol{x}_t)\log \frac{p_{\boldsymbol{\theta}}(\boldsymbol{x}_t \mid \boldsymbol{x}_{t+1})}{\tilde{q}(\boldsymbol{x}_{t+1} \mid \boldsymbol{x}_t)}d\boldsymbol{x}_{t+1}\right)d\boldsymbol{x}_t$$

$$\quad -\int q(\boldsymbol{x}_t)\left(\int\tilde{q}(\boldsymbol{x}_{t+1} \mid \boldsymbol{x}_t)\log \frac{q(\boldsymbol{x}_{t+1})}{\tilde{q}(\boldsymbol{x}_{t+1} \mid \boldsymbol{x}_t)}d\boldsymbol{x}_{t+1}\right)d\boldsymbol{x}_t$$

(23)

$$= -\int \tilde{q}(\boldsymbol{x}_t, \boldsymbol{x}_{t+1})\log \frac{p_{\boldsymbol{\theta}}(\boldsymbol{x}_t \mid \boldsymbol{x}_{t+1})}{\tilde{q}(\boldsymbol{x}_{t+1} \mid \boldsymbol{x}_t)}d\boldsymbol{x}_t d\boldsymbol{x}_{t+1} + C_1$$

$$= -\int \tilde{q}(\boldsymbol{x}_t, \boldsymbol{x}_{t+1})\log \frac{p_{\boldsymbol{\theta}}(\boldsymbol{x}_t \mid \boldsymbol{x}_{t+1})}{\tilde{q}(\boldsymbol{x}_t \mid \boldsymbol{x}_{t+1})} \cdot \frac{q(\boldsymbol{x}_t)}{\tilde{q}(\boldsymbol{x}_{t+1})}d\boldsymbol{x}_t d\boldsymbol{x}_{t+1} + C_1$$

$$= -\int \tilde{q}(\boldsymbol{x}_t, \boldsymbol{x}_{t+1})\log \frac{p_{\boldsymbol{\theta}}(\boldsymbol{x}_t \mid \boldsymbol{x}_{t+1})}{\tilde{q}(\boldsymbol{x}_t \mid \boldsymbol{x}_{t+1})}d\boldsymbol{x}_t d\boldsymbol{x}_{t+1} + C_1 + C_2$$

$$= D_{KL}(\tilde{q}(\boldsymbol{x}_t \mid \boldsymbol{x}_{t+1}) \parallel p_{\boldsymbol{\theta}}(\boldsymbol{x}_t \mid \boldsymbol{x}_{t+1})) + C$$

$$= L_{\text{vlb}}^{\tilde{q}}(\boldsymbol{\theta}, t) + C,$$

where $C, C_1, C_2$ are all constants independent of $\boldsymbol{\theta}$. $\qquad\square$

### A.2.1 Proof of Theorem 1

In this section, we prove the Theorem 1. To simplify the notation, let $p_{\boldsymbol{\theta}}(\boldsymbol{x}_t \mid \boldsymbol{x}_{t+1}) \sim \mathcal{N}(\boldsymbol{\mu}_{\boldsymbol{\theta}}(\boldsymbol{x}_{t+1}, t+1), \sigma_{t+1}$ [6] in (6), then the optimal solution (Lemma 9 in (Bao et al., 2022)) of minimizing $L_{t+1}^{\tilde{q}_t}$ is

$$\boldsymbol{\mu}_{\boldsymbol{\theta}}(\boldsymbol{x}_{t+1}, t+1) = \mathbb{E}_{\tilde{q}_t}[\boldsymbol{x}_t \mid \boldsymbol{x}_{t+1}]. \tag{24}$$

---

[6]Here $\sigma_{t+1}$ can be also optimized as in (Bao et al., 2022), but we find optimizing it in practice does not improve the empirical results.

For every specific $t$, we consider the following $\tilde{q}_t$ in (12) [7], such that

$$
\begin{aligned}
&\tilde{q}_t(\boldsymbol{x}_{t+1} \mid \boldsymbol{x}_t) \neq q(\boldsymbol{x}_{t+1} \mid \boldsymbol{x}_t); \\
&\tilde{q}_t(\boldsymbol{x}_{t+1}) \neq q(\boldsymbol{x}_{t+1}); \\
&\tilde{q}_t(\boldsymbol{x}_{0:t}) = q(\boldsymbol{x}_{0:t}). \\
&\tilde{q}_t(\boldsymbol{x}_t \mid \boldsymbol{x}_0, \boldsymbol{x}_{t+1}) = q(\boldsymbol{x}_t \mid \boldsymbol{x}_0, \boldsymbol{x}_{t+1}) = \mathcal{N}(\mu_{t+1}(\boldsymbol{x}_0, \boldsymbol{x}_{t+1}), \sigma_t).
\end{aligned}
\tag{25}
$$

where $\mu_{t+1}(\boldsymbol{x}_0, \boldsymbol{x}_{t+1}) = \frac{\sqrt{\bar{\alpha}_t}(1-\alpha_{t+1})}{1-\bar{\alpha}_{t+1}}\boldsymbol{x}_0 + \frac{\sqrt{\alpha_{t+1}}(1-\bar{\alpha}_t)}{1-\bar{\alpha}_{t+1}}\boldsymbol{x}_{t+1}$. The $\tilde{q}_t$ can be taken due to the Bayesian rule. Next, we analyze the optimal formulation in (24). Due to the property of conditional expectation, we have

$$
\boldsymbol{\mu}_{\boldsymbol{\theta}}(\boldsymbol{x}_{t+1}, t+1) = \mathbb{E}_{\tilde{q}_t}\left[\mathbb{E}_{\tilde{q}_t}[\boldsymbol{x}_t \mid \boldsymbol{x}_0, \boldsymbol{x}_{t+1}] \mid \boldsymbol{x}_{t+1}\right] = \mu_{t+1}\left(\mathbb{E}_{\tilde{q}_t}[\boldsymbol{x}_0 \mid \boldsymbol{x}_{t+1}], \boldsymbol{x}_{t+1}\right). \tag{26}
$$

As can be seen, the optimal transition rule is decided by the conditional expectation $\mathbb{E}_{\tilde{q}_t}[\boldsymbol{x}_0 \mid \boldsymbol{x}_{t+1}]$ for some $\tilde{q}_t(\boldsymbol{x}_{t+1}) \in B_{D_{KL}}(\tilde{q}(\boldsymbol{x}_{t+1}), \eta_0)$ in (12). Then, we have the following lemma to get the desired conditional expectation.

**Lemma 1.** *There exists some $\eta \geq \eta_0$ in (27) which makes (27) equivalent to problem (12).*

$$
\min_{\boldsymbol{\theta}} \sum_{t=0}^{T-1} \mathbb{E}_{\tilde{q}_t(\boldsymbol{x}_0)} \sup_{\tilde{q}_t(\boldsymbol{x}_{t+1}|\boldsymbol{x}_0) \in B_{D_{KL}}(q_t(\boldsymbol{x}_{t+1}|\boldsymbol{x}_0), \eta)} \mathbb{E}_{\tilde{q}_t(\boldsymbol{x}_{t+1}|\boldsymbol{x}_0)}\left[\|\boldsymbol{x}_{\boldsymbol{\theta}}(\boldsymbol{x}_{t+1}, t+1) - \boldsymbol{x}_0\|^2\right], \tag{27}
$$

*where $\mathbb{E}_{p_{\boldsymbol{\theta}}}[\boldsymbol{x}_0 \mid \boldsymbol{x}_{t+1}] = \boldsymbol{x}_{\boldsymbol{\theta}}(\boldsymbol{x}_{t+1}, t+1)$.*

*Proof.* Let us check the training objective $\min_{\boldsymbol{\theta}} \sup_{\tilde{q}_t \in B_{D_{KL}}(q_{t+1}, \eta)} D_{KL}(\tilde{q}_t(\boldsymbol{x}_t \mid \boldsymbol{x}_{t+1}) \parallel p_{\boldsymbol{\theta}}(\boldsymbol{x}_t \mid \boldsymbol{x}_{t+1}))$. During this proof, we abbreviate $B_{D_{KL}}(q_{t+1}(\boldsymbol{x}_{t+1}), \eta)$ as $B$. Since $p_{\boldsymbol{\theta}}(\boldsymbol{x}_t \mid \boldsymbol{x}_{t+1}) \sim \mathcal{N}(\boldsymbol{\mu}_{\boldsymbol{\theta}}(\boldsymbol{x}_{t+1}, t+1), \sigma_{t+1})$, then

$$
\begin{aligned}
&\sup_{\tilde{q}_t(\boldsymbol{x}_{t+1}) \in B} D_{KL}(\tilde{q}_t(\boldsymbol{x}_t \mid \boldsymbol{x}_{t+1}) \parallel p_{\boldsymbol{\theta}}(\boldsymbol{x}_t \mid \boldsymbol{x}_{t+1})) \\
&\propto -\frac{d}{2}\log 2\pi\sigma_{t+1}^2 - \frac{1}{2\sigma_{t+1}^2} \sup_{\tilde{q}_t(\boldsymbol{x}_{t+1}) \in B} \mathbb{E}_{\tilde{q}(\boldsymbol{x}_t, \boldsymbol{x}_{t+1})}\left[\|\boldsymbol{x}_t - \boldsymbol{\mu}_{\boldsymbol{\theta}}(\boldsymbol{x}_{t+1}, t+1)\|^2\right].
\end{aligned}
\tag{28}
$$

As we consider $\sigma_{t+1}$ as constant, an analysis of the expectation term is enough. Due to

$$
\begin{aligned}
\mathbb{E}_{\tilde{q}_t(\boldsymbol{x}_t, \boldsymbol{x}_{t+1})}\left[\|\boldsymbol{x}_t - \boldsymbol{\mu}_{\boldsymbol{\theta}}(\boldsymbol{x}_{t+1}, t+1)\|^2\right] &\geq \inf_f \mathbb{E}_{\tilde{q}_t(\boldsymbol{x}_0, \boldsymbol{x}_t, \boldsymbol{x}_{t+1})}\left[\|\boldsymbol{x}_t - f(\boldsymbol{x}_0, \boldsymbol{x}_{t+1})\|^2\right] \\
&= \mathbb{E}_{\tilde{q}_t(\boldsymbol{x}_0, \boldsymbol{x}_t, \boldsymbol{x}_{t+1})}\left[\|\boldsymbol{x}_t - \mathbb{E}_{\tilde{q}}[\boldsymbol{x}_t \mid \boldsymbol{x}_0, \boldsymbol{x}_{t+1}]\|^2\right],
\end{aligned}
\tag{29}
$$

where the last term is invariant over $\tilde{q}_t \in B$ so that it is a uniform lower bound over all possible $\tilde{q}_t$ and $p_{\boldsymbol{\theta}}(\boldsymbol{x}_t \mid \boldsymbol{x}_{t+1})$. The above inequality indicates that the optimal $\boldsymbol{\mu}_{\boldsymbol{\theta}}(\boldsymbol{x}_{t+1}, t+1)$ is achieved when the left in (29) becomes the right in (29).

On the other hand, for any $\tilde{q}_t \in B$, let us compute the gap such that

$$
\begin{aligned}
&\mathbb{E}_{\tilde{q}_t(\boldsymbol{x}_t, \boldsymbol{x}_{t+1})}\left[\|\boldsymbol{x}_t - \boldsymbol{\mu}_{\boldsymbol{\theta}}(\boldsymbol{x}_{t+1}, t+1)\|^2\right] \\
&= \mathbb{E}_{\tilde{q}_t}\left[\|\boldsymbol{x}_t - \mathbb{E}_{\tilde{q}_t}[\boldsymbol{x}_t \mid \boldsymbol{x}_0, \boldsymbol{x}_{t+1}] + \mathbb{E}_{\tilde{q}_t}[\boldsymbol{x}_t \mid \boldsymbol{x}_0, \boldsymbol{x}_{t+1}] - \boldsymbol{\mu}_{\boldsymbol{\theta}}(\boldsymbol{x}_{t+1}, t+1)\|^2\right] \\
&= \mathbb{E}_{\tilde{q}_t}\left[\|\boldsymbol{x}_t - \mathbb{E}_{\tilde{q}_t}[\boldsymbol{x}_t \mid \boldsymbol{x}_0, \boldsymbol{x}_{t+1}]\|^2\right] \\
&\quad + \mathbb{E}_{\tilde{q}_t}\left[\|\boldsymbol{\mu}_{\boldsymbol{\theta}}(\boldsymbol{x}_{t+1}, t+1) - \mathbb{E}_{\tilde{q}_t}[\boldsymbol{x}_t \mid \boldsymbol{x}_0, \boldsymbol{x}_{t+1}]\|^2\right] \\
&\quad - 2\mathbb{E}_{\tilde{q}_t}\left[\langle \boldsymbol{x}_t - \mathbb{E}_{\tilde{q}_t}[\boldsymbol{x}_t \mid \boldsymbol{x}_0, \boldsymbol{x}_{t+1}], \boldsymbol{\mu}_{\boldsymbol{\theta}}(\boldsymbol{x}_{t+1}, t+1) - \mathbb{E}_{\tilde{q}_t}[\boldsymbol{x}_t \mid \boldsymbol{x}_0, \boldsymbol{x}_{t+1}]\rangle\right] \\
&= \mathbb{E}_{\tilde{q}_t}\left[\|\boldsymbol{x}_t - \mathbb{E}_{\tilde{q}_t}[\boldsymbol{x}_t \mid \boldsymbol{x}_0, \boldsymbol{x}_{t+1}]\|^2\right] \\
&\quad + \left(\sqrt{\bar{\alpha}_t} - \sqrt{1 - \bar{\alpha}_t - \sigma_{t+1}^2}\sqrt{\frac{\bar{\alpha}_{t+1}}{1 - \bar{\alpha}_{t+1}}}\right)\mathbb{E}_{\tilde{q}_t(\boldsymbol{x}_0, \boldsymbol{x}_{t+1})}\left[\|\boldsymbol{x}_0 - \boldsymbol{x}_{\boldsymbol{\theta}}(\boldsymbol{x}_{t+1}, t+1)\|^2\right],
\end{aligned}
\tag{30}
$$

where the equality is due to the property of conditional expectation leads to $\mathbb{E}_{\tilde{q}_t}[\langle \boldsymbol{x}_t - \mathbb{E}_{\tilde{q}_t}[\boldsymbol{x}_t \mid \boldsymbol{x}_0, \boldsymbol{x}_{t+1}], \boldsymbol{\mu}_{\boldsymbol{\theta}}(\boldsymbol{x}_{t+1}, t+1) - \mathbb{E}_{\tilde{q}_t}[\boldsymbol{x}_t \mid \boldsymbol{x}_0, \boldsymbol{x}_{t+1}]\rangle] = 0$, and rewriting $\mathbb{E}_{\tilde{q}_t}[\|\boldsymbol{\mu}_{\boldsymbol{\theta}}(\boldsymbol{x}_{t+1}, t+1) - \mathbb{E}_{\tilde{q}_t}[\boldsymbol{x}_t \mid \boldsymbol{x}_0, \boldsymbol{x}_{t+1}]\|^2]$ as in equations (5)-(10) in (Ho et al., 2020). Due to this, we know that minimizing the

---

[7]We can do this since (12) only relates to $\tilde{q}_t(\boldsymbol{x}_{t+1})$

square error is equivalent to minimizing the $\mathbb{E}_{\tilde{q}_t(\boldsymbol{x}_t, \boldsymbol{x}_{t+1})}[\|\boldsymbol{x}_0 - \boldsymbol{x}_{\boldsymbol{\theta}}(\boldsymbol{x}_{t+1}, t+1)\|^2]$. On the other hand, since $\tilde{q}_t^* \in B$, then we have

$$
\begin{aligned}
&D_{KL}(q(\boldsymbol{x}_{t+1} \mid \boldsymbol{x}_0) \parallel \tilde{q}_t^*(\boldsymbol{x}_{t+1} \mid \boldsymbol{x}_0)) \\
={}&D_{KL}(q(\boldsymbol{x}_0 \mid \boldsymbol{x}_{t+1}) \parallel \tilde{q}_t^*(\boldsymbol{x}_0 \mid \boldsymbol{x}_{t+1})) + D_{KL}(q(\boldsymbol{x}_{t+1}) \parallel \tilde{q}_t^*(\boldsymbol{x}_{t+1})) \\
\geq{}&\eta_0.
\end{aligned}
\tag{31}
$$

Thus, we prove our conclusion. $\qquad\square$

**Theorem 1.** *There exists $\boldsymbol{\delta}_t$ depends on $\boldsymbol{x}_0$ and $\boldsymbol{\epsilon}_t$ makes* (13) *equivalent to problem* (12).

$$
\min_{\boldsymbol{\theta}} \sum_{t=0}^{T-1} \mathbb{E}_{q(\boldsymbol{x}_0), \boldsymbol{\epsilon}_t} \left[ \left\| \boldsymbol{\epsilon}_{\boldsymbol{\theta}}(\sqrt{\bar{\alpha}_t}\boldsymbol{x}_0 + \sqrt{1-\bar{\alpha}_t}\boldsymbol{\epsilon}_t + \boldsymbol{\delta}_t, t) - \boldsymbol{\epsilon}_t - \frac{\boldsymbol{\delta}_t}{\sqrt{1-\bar{\alpha}_t}} \right\|^2 \right].
\tag{13}
$$

*Proof.* By combining Lemma 1, suppose the supreme is attained under $\tilde{q}_{t-1}$ such that $\boldsymbol{x}_t \sim \tilde{q}_{t-1}(\boldsymbol{x}_t)$ with

$$
\boldsymbol{x}_t = \sqrt{\bar{\alpha}_t}\boldsymbol{x}_0 + \sqrt{1-\bar{\alpha}_t}\boldsymbol{\epsilon}_t + \boldsymbol{\delta}_t,
\tag{32}
$$

with $\boldsymbol{\delta}_t$ depends on $\boldsymbol{x}_0$ and $\boldsymbol{x}_t$. Then we prove the conclusion. $\qquad\square$

### A.2.2 PROOF OF PROPOSITION 5

**Proposition 5.** *If $L_t^{\mathrm{DRO}}(\boldsymbol{\theta}) \leq \eta_0$ in* (12) *for all $t$, and $D_{KL}(q(\boldsymbol{x}_T) \parallel p_{\boldsymbol{\theta}}(\boldsymbol{x}_T)) \leq \eta_0$, then $D_{KL}(q(x_0) \parallel p_{\boldsymbol{\theta}}(\boldsymbol{x}_0)) \leq \eta_0$.*

*Proof.* This theorem can proved by induction. Since $D_{KL}(q(\boldsymbol{x}_T) \parallel p_{\boldsymbol{\theta}}(\boldsymbol{x}_T)) \leq \eta_0$, then, let $\tilde{q}_{T-1}^*(\boldsymbol{x}_T) = p_{\boldsymbol{\theta}}(\boldsymbol{x}_T)$ and satisfies $\tilde{q}_{T-1}^*(\boldsymbol{x}_T) = q(\boldsymbol{x}_{T-1})$. The existence of such distribution is due to Kolmogorov existence theorem (Shiryaev, 2016). Then, we have

$$
\begin{aligned}
D_{KL}(\tilde{q}_{T-1}^*(\boldsymbol{x}_{T-1}) \parallel p_{\boldsymbol{\theta}}(\boldsymbol{x}_{T-1})) \leq{}& D_{KL}(\tilde{q}_{T-1}^*(\boldsymbol{x}_T) \parallel p_{\boldsymbol{\theta}}(\boldsymbol{x}_T)) \\
&+ D_{KL}(\tilde{q}_{T-1}^*(\boldsymbol{x}_{T-1} \mid \boldsymbol{x}_T) \parallel p_{\boldsymbol{\theta}}(\boldsymbol{x}_{T-1} \mid \boldsymbol{x}_T)) \\
\leq{}& L_t^{\mathrm{DRO}}(\boldsymbol{\theta}) \\
\leq{}& \eta_0,
\end{aligned}
\tag{33}
$$

where the first inequality is due to the definition of $L_t^{\mathrm{DRO}}(\boldsymbol{\theta})$ and $\tilde{q}_{T-1}^*(\boldsymbol{x}_T) = p_{\boldsymbol{\theta}}(\boldsymbol{x}_T)$. Then, we prove our conclusion by induction over $t$. $\qquad\square$

### A.2.3 PROOF OF PROPOSITION 6

**Proposition 6.** *For $\eta > 0$ and $\boldsymbol{\delta}_t$ in* (13), *$\|\boldsymbol{\delta}_t\|_1 \leq \eta$ holds with probability at least $1 - \sqrt{2(1-\bar{\alpha}_t)/\eta}$.*

*Proof.* Due to the definition of the first order Wasserstein distance $\mathsf{W}_1(\cdot, \cdot)$ (Villani et al., 2009) for any specific $\boldsymbol{x}_0$, suppose

$$
\pi^* \in \underset{\pi(\boldsymbol{x}_t, \tilde{\boldsymbol{x}}_t) \in q_t(\boldsymbol{x}_t \mid \boldsymbol{x}_0) \times \tilde{q}_t(\tilde{\boldsymbol{x}}_t \mid \boldsymbol{x}_0)}{\arg\min} \mathbb{E}\left[\|\tilde{\boldsymbol{x}}_t - \boldsymbol{x}_t\|_1\right],
\tag{34}
$$

so that

$$
\mathbb{E}_{\pi^*}\left[\|\tilde{\boldsymbol{x}}_t - \boldsymbol{x}_t\|_1\right] = \mathsf{W}_1(q_t(\boldsymbol{x}_t \mid \boldsymbol{x}_0), \tilde{q}_t(\boldsymbol{x}_t \mid \boldsymbol{x}_0)).
\tag{35}
$$

Let $\boldsymbol{\delta}_t$ be the one of (13) under $\pi^*$ derived by Lemma 1, then

$$
\begin{aligned}
\mathbb{P}\left(\|\boldsymbol{\delta}_t\|_1 \geq \eta \mid \boldsymbol{x}_0\right) \leq{}& \frac{\mathbb{E}_{\pi^*}[\|\boldsymbol{\delta}_t\|_1]}{\eta} \\
={}& \frac{\mathsf{W}_1(q_t(\boldsymbol{x}_t \mid \boldsymbol{x}_0), \tilde{q}_t(\boldsymbol{x}_t \mid \boldsymbol{x}_0))}{\eta} \\
\overset{a}{\leq}{}& \frac{\sqrt{2(1-\bar{\alpha}_t)D_{KL}(q_t(\boldsymbol{x}_t \mid \boldsymbol{x}_0) \parallel \tilde{q}_t(\boldsymbol{x}_t \mid \boldsymbol{x}_0))}}{\eta} \\
\leq{}& \sqrt{\frac{2(1-\bar{\alpha}_t)}{\eta}},
\end{aligned}
\tag{36}
$$

where inequality $a$ is due to the Talagrand's inequality (Wainwright, 2019). Then we prove our conclusion. $\qquad\square$

## B    PROOFS IN SECTION 5

Next, we give the proof of results in Section 5. Firstly, let us check the definition of the $\Phi_t(\boldsymbol{x}_{t+1})$. For the variance-preserving stochastic differential equation in Song et al. (2022)

$$d\boldsymbol{z}_s = -\frac{\beta_s}{2}\boldsymbol{z}_s dt + \sqrt{\beta_s}dW_s. \tag{37}$$

Due to the solution of $\boldsymbol{z}_s$ in Song et al. (2023), we know $\boldsymbol{z}_{s_t}$ has the same distribution with $\boldsymbol{x}_t$ in (1) for $\{s_t\}_{t=1}^T$ satisfies

$$\exp\left(-\int_0^{s_t} \beta(u)du\right) = \bar{\alpha}_t \qquad (s_0 = 0). \tag{38}$$

In the rest of this section, we use $d(\boldsymbol{x}, \boldsymbol{y})$ in (15) as $\ell_2$ distance $\|\boldsymbol{x} - \boldsymbol{y}\|^2$, whereas the conclusions under other distance can be similarly derived. Owing the the discussion in above, similar to (Song et al., 2023), when $\boldsymbol{x}_{t+1} = \boldsymbol{z}_{s_{t+1}}$, let $\Phi_t(\boldsymbol{x}_{t+1}) = \Psi_{s_t}(\boldsymbol{z}_{s_{t+1}})$, we can rewrite the objective (15) as follows.

$$\min_{\boldsymbol{\theta}} \mathcal{L}_{CD}(\boldsymbol{\theta}) = \min_{\boldsymbol{\theta}} \sum_{t=0}^{T-1} \mathbb{E}_{\boldsymbol{z}_{s_t}} \left[\left\|f_{\boldsymbol{\theta}}(\Psi_{s_t}(\boldsymbol{z}_{s_{t+1}}), t) - f_{\boldsymbol{\theta}}(\boldsymbol{z}_{s_{t+1}}, t+1)\right\|^2\right]. \tag{39}$$

Here $\boldsymbol{z}_s$ follows the following reverse time ODE of (37) with $\boldsymbol{z}_0 \sim q(\boldsymbol{x}_0)$,

$$d\boldsymbol{z}_s = \underbrace{-\frac{\beta_s}{2}\left(\boldsymbol{z}_s + \frac{1}{2}\nabla_{\boldsymbol{z}} \log q_s(\boldsymbol{z}_s)\right) ds}_{\phi_s}, \tag{40}$$

and such $\boldsymbol{z}_s$ has the same distribution with the ones in (37) (Song et al., 2022), where $q_s$ is the density of $\boldsymbol{z}_s$. $\Psi_{s_t}(\boldsymbol{z}_{s_{t+1}}) = \boldsymbol{z}_{s_{t+1}} - \int_{s_t}^{s_{t+1}} \phi_s(\boldsymbol{z}_s)ds$, which is a deterministic function of $\boldsymbol{z}_{s_{t+1}}$, and $f_{\boldsymbol{\theta}}(\boldsymbol{z}_{s_0}, 0) = \boldsymbol{z}_{s_0} = \boldsymbol{z}_0$.

Now, we are ready to prove the Theorem 2 as follows.

**Theorem 2.** *For $\mathcal{L}_{CD}(\boldsymbol{\theta})$ in (15) with $d(\cdot, \cdot)$ is $\ell_2$ distance, then $\mathsf{W}_1(f_{\boldsymbol{\theta}}(\boldsymbol{x}_t, t), \boldsymbol{x}_0) \leq \sqrt{t\mathcal{L}_{CD}(\boldsymbol{\theta})}$ [8].*

*Proof.* Owing to the definition of $\mathsf{W}_1$-distance, and the discussion in above, we have

$$\begin{aligned}
\mathsf{W}_1(f_{\boldsymbol{\theta}}(\boldsymbol{x}_T, T), \boldsymbol{x}_0) &= \mathsf{W}_1(f_{\boldsymbol{\theta}}(\boldsymbol{z}_{s_T}, T), \boldsymbol{z}_{s_0}) \\
&= \mathsf{W}_1\left(f_{\boldsymbol{\theta}}(\boldsymbol{z}_{s_T}, T), \Psi_{s_0}\left(\Psi_{s_1}\left(\cdots \Psi_{s_{T-1}}\left(\boldsymbol{z}_{s_T}\right)\right)\right)\right) \\
&\leq \mathbb{E}\left[\left\|f_{\boldsymbol{\theta}}(\boldsymbol{z}_{s_T}, T) - \Psi_{s_0}\left(\Psi_{s_1}\left(\cdots \Psi_{s_{T-1}}\left(\boldsymbol{z}_{s_T}\right)\right)\right)\right\|\right] \\
&\leq \sum_{t=0}^{T-1} \mathbb{E}\left[\left\|f_{\boldsymbol{\theta}}(\boldsymbol{z}_{s_{t+1}}, t+1) - f_{\boldsymbol{\theta}}(\Psi_{s_t}(\boldsymbol{z}_{s_{t+1}}), t)\right\|\right] \\
&\leq \sqrt{T\mathcal{L}_{CD}(\boldsymbol{\theta})},
\end{aligned} \tag{41}$$

where the first inequality is due to the definition of Wasserstein distance, the second and last inequalities respectively use the triangle inequality and Schwarz's inequality. $\square$

### B.1    PROOF OF THEOREM 3

As pointed out in the above, the used $\hat{\Phi}_t(\boldsymbol{x}_{t+1}, \boldsymbol{\epsilon}_\phi)$ is a numerical estimator of $\Phi_t(\boldsymbol{x}_{t+1})$. In the sequel, let us consider $\hat{\Phi}$ is an Euler estimator as follows, whereas our analysis can be similarly generalized to the other estimators.

$$\hat{\Phi}_t(\boldsymbol{x}_{t+1}, \boldsymbol{\epsilon}_\phi) = \hat{\Psi}_{s_t}(\boldsymbol{z}_{s_{t+1}}, \boldsymbol{\epsilon}_\phi) = \boldsymbol{z}_{s_{t+1}} + (s_{t+1} - s_t)\underbrace{\frac{\beta_{s_{t+1}}}{2}\left(\boldsymbol{z}_{s_{t+1}} + \boldsymbol{\epsilon}_\phi(\boldsymbol{z}_{s_{t+1}}, t+1)/\sqrt{1 - \bar{\alpha}_{t+1}}\right)}_{\hat{\phi}_{s_{t+1}}},$$
$$\tag{42}$$

---

[8] Here $\mathsf{W}_1(f_{\boldsymbol{\theta}}(\boldsymbol{x}_t, t), \boldsymbol{x}_0)$ is the Wasserstein 1-distance between distributions of $f_{\boldsymbol{\theta}}(\boldsymbol{x}_t, t)$ and $\boldsymbol{x}_0$.

where $\sqrt{1 - \bar{\alpha}_{t+1}}\boldsymbol{\epsilon}_{\boldsymbol{\phi}}(\boldsymbol{z}_{s_{t+1}}, t+1)$ estimates $\nabla_{\boldsymbol{z}} \log q_{s_{t+1}}(\boldsymbol{z}_{s_{t+1}})$ as pointed out in (Song et al., 2020), and the condition $\boldsymbol{x}_{t+1} = \boldsymbol{z}_{s_{t+1}}$ is hold.

Next, we illustrate the used regularity conditions to derive Theorem 3.

**Assumption 1.** *The discretion error of $\hat{\Psi}_{s_t}(\boldsymbol{z}_{s_{t+1}}, \boldsymbol{\epsilon}_{\boldsymbol{\phi}})$ is smaller than $C(s_{t+1} - s_t)^2$ for constant $C$, that says*

$$\left\| \hat{\Psi}_{s_t}(\boldsymbol{z}_{s_{t+1}}, \boldsymbol{\epsilon}_{\boldsymbol{\phi}}) - \boldsymbol{z}_{s_{t+1}} - \int_{s_t}^{s_{t+1}} \hat{\phi}_s(\boldsymbol{z}_s) ds \right\| \leq C(s_{t+1} - s_t)^2 \tag{43}$$

**Assumption 2.** *The estimated score $\nabla_{\boldsymbol{z}} \log \hat{q}_s(\boldsymbol{z})$ has bounded expected error, i.e.,*

$$\mathbb{E}_{\boldsymbol{z} \sim q_{s_t}(\boldsymbol{z})} \left[ \left\| \hat{\phi}_{s_t}(\boldsymbol{z}) - \phi_{s_t}(\boldsymbol{z}) \right\|^2 \right] \leq \epsilon. \tag{44}$$

*for all $0 \leq t < T$.*

**Assumption 3.** *For the learned model $f_{\boldsymbol{\theta}}$, it holds $\|f_{\boldsymbol{\theta}}\| \leq D$.*

The Assumption 1 describes the discretion error of the Euler method under ODE with drift term $\hat{\phi}_s$, which can be satisfied under proper continuity conditions of model $\boldsymbol{\epsilon}_{\boldsymbol{\phi}}$. On the other hand, Assumption 2 describes the estimation error of $\hat{\phi}_{s_t}(\boldsymbol{z})$, which terms out to be the training objective of obtaining it, see (Song et al., 2020) for more details. The Assumption 3 is natural, since $f_{\boldsymbol{\theta}}$ predicts $\boldsymbol{x}_0$, which is usually an image data with bounded norm. Now, we are ready to prove the Theorem 3, which is presented by proving the following formal version.

**Theorem 4.** *Under Assumptions 1, 2, and 3, for all $\boldsymbol{\delta}_{s_t}$, we have $\mathbb{E}_{\boldsymbol{z}_{s_t}}[\|\boldsymbol{\delta}_{s_t}(\boldsymbol{z}_{s_t})\|] \leq O(\Delta_{s_t}^2 + \epsilon\sqrt{\Delta_{s_t}})$ for $\Delta_{s_t} = s_{t+1} - s_t$. Besides that, we have*

$$\mathsf{W}_1(f_{\boldsymbol{\theta}}(\boldsymbol{z}_T, T), \boldsymbol{z}_0) \leq \sqrt{T\hat{\mathcal{L}}_{CD}^{Adv}(\boldsymbol{\theta}) + \frac{4D^2}{\eta}\left[ C\Delta_{s_t}^2 + \epsilon O(\sqrt{\Delta_{s_t}}) \right]}. \tag{45}$$

*Proof.* Noting that $\Phi_t(\boldsymbol{x}_{t+1}) = \Psi_{s_t}(\boldsymbol{z}_{s_{t+1}})$ and $\hat{\Phi}_t(\boldsymbol{x}_{t+1}, \boldsymbol{\epsilon}_{\boldsymbol{\phi}}) = \hat{\Psi}_{s_t}(\boldsymbol{z}_{s_{t+1}}, \boldsymbol{\epsilon}_{\boldsymbol{\phi}})$, the key problem is to upper bound the difference between $\hat{\Psi}_{s_t}(\boldsymbol{z}, \boldsymbol{\epsilon}_{\boldsymbol{\phi}})$ and $\Psi_{s_t}(\boldsymbol{z})$ for all $t$ and $\boldsymbol{z}$. To do so, we note that

$$\left\| \hat{\Psi}_{s_t}(\boldsymbol{z}, \boldsymbol{\epsilon}_{\boldsymbol{\phi}}) - \Psi_{s_t}(\boldsymbol{z}) \right\| \leq \left\| \hat{\Psi}_{s_t}(\boldsymbol{z}, \boldsymbol{\epsilon}_{\boldsymbol{\phi}}) - \boldsymbol{z} - \int_{s_t}^{s_{t+1}} \hat{\phi}_s(\boldsymbol{z}_s) ds \right\| + \left\| \boldsymbol{z} - \int_{s_t}^{s_{t+1}} \hat{\phi}_s(\boldsymbol{z}_s) ds - \Psi_{s_t}(\boldsymbol{z}) \right\|, \tag{46}$$

where the first one in r.h.s can be upper bounded by $C(s_{t+1} - s_t)^2$ according to Assumption 1. On the other hand, define $\frac{d\hat{\boldsymbol{z}}_s}{ds} = \hat{\phi}_s(\hat{\boldsymbol{z}}_s)$, then when $\hat{\boldsymbol{z}}_{s_{t+1}} = \boldsymbol{z}_{s_{t+1}} = \boldsymbol{z}$ and $s \in [s_t, s_{t+1}]$.

$$\begin{aligned}
\frac{d}{ds}\|\hat{\boldsymbol{z}}_s - \boldsymbol{z}_s\|^2 &= \left\langle \hat{\boldsymbol{z}}_s - \boldsymbol{z}_s, \hat{\phi}_s(\hat{\boldsymbol{z}}_s) - \phi_s(\boldsymbol{z}_s) \right\rangle \\
&= \left\langle \hat{\boldsymbol{z}}_s - \boldsymbol{z}_s, \hat{\phi}_s(\hat{\boldsymbol{z}}_s) - \hat{\phi}_s(\boldsymbol{z}_s) + \hat{\phi}_s(\boldsymbol{z}_s) - \phi_s(\boldsymbol{z}_s) \right\rangle \\
&\leq L\|\hat{\boldsymbol{z}}_s - \boldsymbol{z}_s\|^2 + \left\langle \hat{\boldsymbol{z}}_s - \boldsymbol{z}_s, \hat{\phi}_s(\boldsymbol{z}_s) - \phi_s(\boldsymbol{z}_s) \right\rangle \\
&\leq \left(\frac{1}{2} + L\right)\|\hat{\boldsymbol{z}}_s - \boldsymbol{z}_s\|^2 + \frac{1}{2}\left\| \hat{\phi}_s(\boldsymbol{z}_s) - \phi_s(\boldsymbol{z}_s) \right\|^2.
\end{aligned} \tag{47}$$

Taking expectation over $\boldsymbol{z}$, by Gronwall's inequality, Assumption 2 and $\hat{\boldsymbol{z}}_{s_{t+1}} = \boldsymbol{z}_{s_{t+1}}$, we have

$$\mathbb{E}\left[\|\hat{\boldsymbol{z}}_{s_t} - \boldsymbol{z}_{s_t}\|^2\right] \leq \int_{s_t}^{s_{t+1}} \frac{e^{(1/2+L)(s-s_t)}}{2} \mathbb{E}\left[\|\hat{\phi}_s(\boldsymbol{z}_s) - \phi_s(\boldsymbol{z}_s)\|^2\right] ds \leq \frac{\epsilon}{4} \int_{s_t}^{s_{t+1}} \beta_s e^{(1/2+L)(s-s_t)} ds. \tag{48}$$

Plugging this into (46), we know

$$\mathbb{E}\left[\left\| \hat{\Psi}_{s_t}(\boldsymbol{z}_{s_t}, \boldsymbol{\epsilon}_{\boldsymbol{\phi}}) - \Psi_{s_t}(\boldsymbol{z}_{s_t}) \right\|\right] \leq C(s_{t+1} - s_t)^2 + \epsilon O(\sqrt{s_{t+1} - s_t}). \tag{49}$$

By Markov's inequality, we have

$$\begin{aligned}
\mathbb{P}\left(\left\| \hat{\Psi}_{s_t}(\boldsymbol{z}_{s_t}, \boldsymbol{\epsilon}_{\boldsymbol{\phi}}) - \Psi_{s_t}(\boldsymbol{z}_{s_t}) \right\| \geq \eta\right) &\leq \frac{\mathbb{E}\left[\left\| \hat{\Psi}_{s_t}(\boldsymbol{z}_{s_t}, \boldsymbol{\epsilon}_{\boldsymbol{\phi}}) - \Psi_{s_t}(\boldsymbol{z}_{s_t}) \right\|\right]}{\eta} \\
&\leq \frac{1}{\eta}\left[ C(s_{t+1} - s_t)^2 + \epsilon O(\sqrt{s_{t+1} - s_t}) \right].
\end{aligned} \tag{50}$$

Thus,

$$\mathbb{E}\left[\|f_{\boldsymbol{\theta}}(\boldsymbol{x}_{t+1}, t+1) - f_{\boldsymbol{\theta}}(\Phi_t(\boldsymbol{x}_{t+1}), t)\|^2\right]$$

$$= \mathbb{E}\left[\left\|f_{\boldsymbol{\theta}}(\boldsymbol{z}_{s_{t+1}}, t+1) - f_{\boldsymbol{\theta}}(\Psi_{s_t}(\boldsymbol{z}_{s_{t+1}}), t)\right\|^2\right]$$

$$= \mathbb{E}\left[\left\|f_{\boldsymbol{\theta}}(\boldsymbol{z}_{s_{t+1}}, t+1) - f_{\boldsymbol{\theta}}(\hat{\Psi}_{s_t}(\boldsymbol{z}_{s_{t+1}} + \boldsymbol{\delta}_{s_t}, \boldsymbol{\epsilon}_\phi), t)\right\|\right]$$

$$= \mathbb{E}\left[\left(\mathbf{1}_{\|\delta_{s_t}\|>\eta} + \mathbf{1}_{\|\delta_{s_t}\|\le\eta}\right)\left\|f_{\boldsymbol{\theta}}(\boldsymbol{z}_{s_{t+1}}, t+1) - f_{\boldsymbol{\theta}}(\hat{\Psi}_{s_t}(\boldsymbol{z}_{s_{t+1}} + \boldsymbol{\delta}_{s_t}, \boldsymbol{\epsilon}_\phi), t)\right\|^2\right]$$

$$\le \mathbb{E}\left[\sup_{\|\boldsymbol{\delta}\|\le\eta}\left\|f_{\boldsymbol{\theta}}(\boldsymbol{z}_{s_{t+1}}, t+1) - f_{\boldsymbol{\theta}}(\hat{\Psi}_{s_t}(\boldsymbol{z}_{s_{t+1}} + \boldsymbol{\delta}_{s_t}, \boldsymbol{\epsilon}_\phi), t)\right\|\right] + 4D^2\mathbb{P}\left(\|\delta_{s_t}\|^2 \ge \eta\right)$$

$$\le \mathbb{E}\left[\sup_{\|\boldsymbol{\delta}\|\le\eta}\left\|f_{\boldsymbol{\theta}}(\boldsymbol{z}_{s_{t+1}}, t+1) - f_{\boldsymbol{\theta}}(\hat{\Psi}_{s_t}(\boldsymbol{z}_{s_{t+1}} + \boldsymbol{\delta}, \boldsymbol{\epsilon}_\delta), t)\right\|^2\right] + \frac{4D^2}{\eta}\left[C(s_{t+1} - s_t)^2 + \epsilon O(\sqrt{s_{t+1} - s_t})\right]. \tag{51}$$

Taking sum over $t$ and combining Theorem 2, we prove our conclusion. $\qquad\square$

Therefore, in this theorem, by taking $\Delta_{s_t} = s_{t+1} - s_t$ close to zero, we get the results in Theorem 3.

## C  THE CONNECTION TO STANDARD ADVERSARIAL TRAINING

In this section, we clarify why the proposed AT objective (14) is a general version of the standard AT objective proposed in (Madry et al., 2018) used for classification problems.

For classification problem, given model $f_{\boldsymbol{\theta}}(\boldsymbol{x})$, data $\boldsymbol{x}$, and label $y$, it aims to minimize the adversarial training objective

$$\min_{\boldsymbol{\theta}} \mathbb{E}_{(\boldsymbol{x},y)}\left[\sup_{\boldsymbol{\delta}:\|\boldsymbol{\delta}\|\le\eta_0} \ell(f_{\boldsymbol{\theta}}(\boldsymbol{x} + \boldsymbol{\delta}), y)\right], \tag{52}$$

for some loss function $\ell$ (e.g. cross entropy) and adversarial radius $\eta_0$. However, the objective is not directly generalized to the diffusion model, as its training objective is a regression problem instead of classification (52). Thus, we should refer to the general version of adversarial training as in (Yi et al., 2021; Sinha et al., 2018), where the training objective is $\min_{\boldsymbol{\theta}} \mathbb{E}_{\boldsymbol{x}}[\ell_{\boldsymbol{\theta}}(\boldsymbol{x})]$, and the adversarial training objective becomes

$$\min_{\boldsymbol{\theta}} \mathbb{E}_{\boldsymbol{x}}\left[\sup_{\boldsymbol{\delta}:\|\boldsymbol{\delta}\|\le\eta_0} \ell_{\boldsymbol{\theta}}(\boldsymbol{x} + \boldsymbol{\delta}))\right], \tag{53}$$

where $\ell_{\boldsymbol{\theta}}$ is the parameterized loss function, and $\boldsymbol{x}$ is data. Then, we can conclude our objective (14) follows the above formulation, such that the goal is represented as

$$\min_{\boldsymbol{\theta}} \sum_{t=0}^{T-1} \mathbb{E}_{\boldsymbol{x}_0}\left[\mathbb{E}_{\boldsymbol{x}_t|\boldsymbol{x}_0}\left[\sup_{\boldsymbol{\delta}:\|\boldsymbol{\delta}\|\le\eta_0} \ell_{\boldsymbol{\theta}}^{\boldsymbol{x}_0}(\boldsymbol{x}_t + \boldsymbol{\delta})\right]\right], \tag{54}$$

compared with the original noise prediction objective $\min_{\boldsymbol{\theta}} \sum_{t=0}^{T-1} \mathbb{E}_{\boldsymbol{x}_0}\left[\mathbb{E}_{\boldsymbol{x}_t|\boldsymbol{x}_0}[\ell_{\boldsymbol{\theta}}^{\boldsymbol{x}_0}(\boldsymbol{x}_t)]\right]$ (5), such that the loss function

$$\ell_{\boldsymbol{\theta}}^{\boldsymbol{x}_0}(\boldsymbol{x}_t) = \left\|\boldsymbol{\epsilon}_{\boldsymbol{\theta}}(t, \boldsymbol{x}_t) - \frac{\boldsymbol{x}_t - \sqrt{\bar{\alpha}_t}\boldsymbol{x}_0}{\sqrt{1 - \bar{\alpha}_t}}\right\|^2. \tag{55}$$

This clarifies the equivalence of our objective (14) to general adversarial training.

## D  ADVERSARIAL TRAINING ON CONSISTENCY TRAINING MODEL

In (Song et al., 2023), the consistency model can be even trained without estimator $\hat{\phi}_s$. They prove that the empirical consistency distillation loss $\hat{\mathcal{L}}_{CD}(\boldsymbol{\theta})$ can be approximated by the following $\mathcal{L}_{CT}(\boldsymbol{\theta})$

$$\mathcal{L}_{CT}(\boldsymbol{\theta}) = \sum_{t=0}^{T-1} \mathbb{E}_{\boldsymbol{x}_{t+1}\sim q(\boldsymbol{x}_{t+1})}\left[\|f_{\boldsymbol{\theta}}(\boldsymbol{x}_t, t) - f_{\boldsymbol{\theta}}(\boldsymbol{x}_{t+1}, t+1)\|^2\right]. \tag{56}$$

In our adversarial regime, we can also prove that the desired $\hat{\mathcal{L}}_{CD}^{Adv}(\boldsymbol{\theta})$ can be approximated by the following $\mathcal{L}_{CT}^{Adv}(\boldsymbol{\theta})$ with adversarial perturbation

$$\mathcal{L}_{CT}^{Adv}(\boldsymbol{\theta}) = \sum_{t=0}^{T-1} \mathbb{E}_{\boldsymbol{x}_{t+1} \sim q(\boldsymbol{x}_{t+1})} \left[ \sup_{\|\boldsymbol{\delta}\| \leq \eta} \|f_{\boldsymbol{\theta}}(\boldsymbol{x}_t + \boldsymbol{\delta}, t) - f_{\boldsymbol{\theta}}(\boldsymbol{x}_{t+1}, t+1)\|^2 \right]. \tag{57}$$

The results can be checked by the following theorem.

**Theorem 5.** *Suppose $f_{\boldsymbol{\theta}}(\boldsymbol{x}_t, t)$ is twice continuously differentiable with a bounded second derivative. Then*

$$\hat{\mathcal{L}}_{CD}^{Adv}(\boldsymbol{\theta}) \lesssim \mathcal{L}_{CT}^{Adv}(\boldsymbol{\theta}) + O\left( T - \sum_{t=1}^{T} \sqrt{\alpha_t} + T\eta^2 \right), \tag{58}$$

*where "$\lesssim$" means approximately less than or equal.*

*Proof.* Due to the continuity of $f_{\boldsymbol{\theta}}(\boldsymbol{x}, t)$, for any $\boldsymbol{\delta}$ with $\|\boldsymbol{\delta}\| \leq \eta$, by Taylor's expansion on $\boldsymbol{x}_{t+1}$ from $\boldsymbol{x}_t + \boldsymbol{\delta}$, we have

$$\mathbb{E}\left[\|f_{\boldsymbol{\theta}}(\boldsymbol{x}_t + \boldsymbol{\delta}, t) - f_{\boldsymbol{\theta}}(\boldsymbol{x}_{t+1}, t+1)\|^2\right] = \mathbb{E}\left[\|f_{\boldsymbol{\theta}}(\boldsymbol{x}_{t+1}, t) - f_{\boldsymbol{\theta}}(\boldsymbol{x}_{t+1}, t+1)\|^2\right]$$
$$+ \mathbb{E}\left[(f_{\boldsymbol{\theta}}(\boldsymbol{x}_{t+1}, t) - f_{\boldsymbol{\theta}}(\boldsymbol{x}_{t+1}, t+1))^\top \nabla f_{\boldsymbol{\theta}}(\boldsymbol{x}_{t+1}, t)(\boldsymbol{x}_t + \boldsymbol{\delta} - \boldsymbol{x}_{t+1})\right] + O\left(\mathbb{E}\left[\|\boldsymbol{x}_{t+1} - \boldsymbol{x}_t - \boldsymbol{\delta}\|^2\right]\right). \tag{59}$$

Due to the Taylor's expansion $f_{\boldsymbol{\theta}}(\boldsymbol{x}_t + \boldsymbol{\delta}, t) = f_{\boldsymbol{\theta}}(\boldsymbol{x}_{t+1}, t) + \nabla f_{\boldsymbol{\theta}}(\boldsymbol{x}_{t+1}, t)(\boldsymbol{x}_t + \boldsymbol{\delta} - \boldsymbol{x}_{t+1}) + \mathcal{O}(\|\boldsymbol{x}_{t+1} - \boldsymbol{x}_t - \boldsymbol{\delta}\|^2)$. Then, from the formulation of $\boldsymbol{x}_t$, we know $\mathbb{E}\left[\|\boldsymbol{x}_{t+1} - \boldsymbol{x}_t - \boldsymbol{\delta}\|^2\right] = O(1 - \sqrt{\alpha_t} + \eta^2)$. Noting that due to definition of $s_t$, we have

$$\mathbb{E}[\boldsymbol{x}_t \mid \boldsymbol{x}_{t+1} = \boldsymbol{z}_{s_{t+1}}] = \mathbb{E}[\boldsymbol{z}_{s_t} \mid \boldsymbol{z}_{s_{t+1}}]$$

$$= \frac{1}{\sqrt{\alpha_{t+1}}} \left( \boldsymbol{z}_{s_{t+1}} - (1 - \alpha_{t+1})\nabla_{\boldsymbol{x}} \log q_{s_{t+1}}(\boldsymbol{z}_{s_{t+1}}) \right)$$

$$= \exp\left( \frac{1}{2} \int_{s_t}^{s_{t+1}} \beta_s ds \right) \left( \boldsymbol{z}_{s_{t+1}} - \left( 1 - e^{\int_{s_t}^{s_{t+1}} \beta_s ds} \right) \nabla_{\boldsymbol{z}} \log q_{s_{t+1}}(\boldsymbol{z}_{s_{t+1}}) \right) \tag{60}$$

$$\approx \left( 1 + \frac{1}{2} \int_{s_t}^{s_{t+1}} \beta_s ds \right) \boldsymbol{z}_{s_{t+1}} + \frac{1}{2} \int_{s_t}^{s_{t+1}} \beta_s ds \nabla_{\boldsymbol{z}} \log q_{s_{t+1}}(\boldsymbol{z}_{s_{t+1}})$$

$$\approx \hat{\Psi}_{s_t}(\boldsymbol{z}_{s_{t+1}}, \sqrt{1 - \bar{\alpha}_{t+1}} \nabla_{\boldsymbol{z}} \log q_{s_{t+1}}),$$

where the first equality is due to Tweedie's formula i.e., Lemma 11 in (Bao et al., 2022), the "$\approx$" is due to $e^a \approx 1 + a$ when $a \to 0$, and the last $\approx$ is due to Euler-Mayaruma discretion. Due to this, we notice that

$$\mathbb{E}\left[(f_{\boldsymbol{\theta}}(\boldsymbol{x}_{t+1}, t) - f_{\boldsymbol{\theta}}(\boldsymbol{x}_{t+1}, t+1))^\top \nabla f_{\boldsymbol{\theta}}(\boldsymbol{x}_{t+1}, t)(\boldsymbol{x}_t + \boldsymbol{\delta} - \boldsymbol{x}_{t+1}) \mid \boldsymbol{x}_{t+1} = \boldsymbol{z}_{s_{t+1}}\right]$$

$$= \mathbb{E}\left[(f_{\boldsymbol{\theta}}(\boldsymbol{x}_{t+1}, t) - f_{\boldsymbol{\theta}}(\boldsymbol{x}_{t+1}, t+1))^\top \nabla f_{\boldsymbol{\theta}}(\boldsymbol{x}_{t+1}, t) \left( \mathbb{E}\left[\boldsymbol{x}_t + \boldsymbol{\delta} \mid \boldsymbol{x}_{t+1} = \boldsymbol{z}_{s_{t+1}}\right] - \boldsymbol{x}_{t+1} \right) \mid \boldsymbol{x}_{t+1} = \boldsymbol{z}_{s_{t+1}}\right]$$

$$\approx \mathbb{E}\left[(f_{\boldsymbol{\theta}}(\boldsymbol{z}_{s_{t+1}}, t) - f_{\boldsymbol{\theta}}(\boldsymbol{z}_{s_{t+1}}, t+1))^\top \nabla f_{\boldsymbol{\theta}}(\boldsymbol{z}_{s_{t+1}}, t) \left( \hat{\Psi}_{s_t}(\boldsymbol{z}_{s_{t+1}}, \nabla_{\boldsymbol{z}} \log q_{s_{t+1}}) + \mathbb{E}[\boldsymbol{\delta} \mid \boldsymbol{z}_{s_{t+1}}] - \boldsymbol{z}_{t+1} \right)\right], \tag{61}$$

where the first equality is due to the property of conditional expectation, and the second "$\approx$" is due to (60). Combining this with (59), we have

$$\mathbb{E}\left[\|f_{\boldsymbol{\theta}}(\boldsymbol{x}_t + \boldsymbol{\delta}, t) - f_{\boldsymbol{\theta}}(\boldsymbol{x}_{t+1}, t+1)\|^2 \mid \boldsymbol{x}_{t+1} = \boldsymbol{z}_{s_{t+1}}\right]$$

$$= \mathbb{E}\left[\|f_{\boldsymbol{\theta}}(\boldsymbol{z}_{s_t} + \boldsymbol{\delta}, t) - f_{\boldsymbol{\theta}}(\boldsymbol{z}_{s_{t+1}}, t+1)\|^2 \mid \boldsymbol{z}_{s_{t+1}}\right]$$

$$= \mathbb{E}\left[\|f_{\boldsymbol{\theta}}(\boldsymbol{z}_{s_{t+1}}, t) - f_{\boldsymbol{\theta}}(\boldsymbol{z}_{s_{t+1}}, t+1)\|^2\right]$$

$$+ \mathbb{E}\left[(f_{\boldsymbol{\theta}}(\boldsymbol{z}_{s_{t+1}}, t) - f_{\boldsymbol{\theta}}(\boldsymbol{z}_{s_{t+1}}, t+1))^\top \nabla f_{\boldsymbol{\theta}}(\boldsymbol{z}_{s_{t+1}}, t) \left( \hat{\Psi}_{s_t}(\boldsymbol{z}_{s_{t+1}}, \nabla_{\boldsymbol{z}} \log q_{s_{t+1}}) + \mathbb{E}[\boldsymbol{\delta} \mid \boldsymbol{z}_{s_{t+1}}] - \boldsymbol{z}_{s_{t+1}} \right)\right]$$

$$+ O(1 - \sqrt{\alpha_t} + \eta^2)$$

$$= \mathbb{E}\left[\|f_{\boldsymbol{\theta}}(\hat{\Psi}_{s_t}(\boldsymbol{z}_{s_{t+1}}, \nabla_{\boldsymbol{z}} \log q_{s_{t+1}}) + \boldsymbol{\delta}, t) - f_{\boldsymbol{\theta}}(\boldsymbol{z}_{s_{t+1}}, t+1)\|^2\right] + O(1 - \sqrt{\alpha_t} + \eta^2), \tag{62}$$

where the last equality is due to Taylor's expansion from $f_{\boldsymbol{\theta}}(\hat{\Psi}_{s_t}(\boldsymbol{z}_{s_{t+1}}, \nabla_{\boldsymbol{z}} \log q_{s_{t+1}}) + \boldsymbol{\delta}, t)$ to $f_{\boldsymbol{\theta}}(\boldsymbol{z}_{s_{t+1}}, t)$. Due to the arbitrariness of $\boldsymbol{\delta}$, we prove our conclusion. $\square$

# E IMPLEMENTATION DETAILS

## E.1 HYPERPARAMETERS OF DIFFUSION MODELS

For the diffusion models, all methods adopt the ADM model (Dhariwal & Nichol, 2021) as the backbone and follow the same training pipeline. Following existing work (Dhariwal & Nichol, 2021; Ning et al., 2023), we train models using the AdamW optimizer (Loshchilov & Hutter, 2019) with mixed precision training and the EMA rate is set to 0.9999. For `CIFAR-10`, the pretrained ADM is trained using a batch size of 128 for 250K iterations with a learning rate set to 1e-4. For `ImageNet`, the pretrained model is trained with a batch size of 1024 for 400K iterations, employing a learning rate of 3e-4. The models are trained in a cluster of NVIDIA Tesla V100s. More hyperparameters are reported in Table 4.

Table 4: Hyperparameters of diffusion model on each datasets.

| Hyperparameters | `CIFAR10` $32 \times 32$ | `ImageNet` $64 \times 64$ |
|---|---|---|
| Channels | 128 | 192 |
| Batch size | 128 | 1024 |
| Learning rate | 1e-4 | 3e-4 |
| Fine-tuning iterations | 200K | 200K |
| Dropout | 0.3 | 0.1 |
| Noise schedule | Cosine | Cosine |

## E.2 HYPERPARAMETERS OF LATENT CONSISTENCY MODELS

For experiments on Latent Consistency Models (LCM) (Luo et al., 2023), we train models on LAIOIN-Aesthetic-6.5+ (Schuhmann et al., 2022) at the resolution of $512 \times 512$, comprising 650K text-image pairs with predicted aesthetic scores higher than 6.5. Stable Diffusion v1.5 (Rombach et al., 2022) is adopted as the teacher model and initialized the student and target models in the latent consistency distillation framework. We set the range of the guidance scale $[w_{min}, w_{max}] = [3, 5]$ during training and use $w = 4$ in sampling because it performs better in our preliminary experiments, which is similar to DMD (Yin et al., 2024). The models are trained in a cluster of NVIDIA Tesla V100s. Both models of our AT and the original LCM training are trained from scratch with the same hyperparameters. We select the adversarial learning rate $\alpha$ from $\{0.02, 0.05\}$ and adversarial step $K$ from $\{2, 3\}$. More details of hyperparameters are shown in Table 5 and other details of implementations can be found in the original LCM paper (Luo et al., 2023).

Table 5: Hyperparameters of latent consistency model.

| Hyperparameters | LAIOIN-Aesthetic-6.5+ |
|---|---|
| Batch size | 64 |
| Learning rate | 8e-6 |
| Training iterations | 100K |
| EMA rate of target model | 0.95 |
| Conditional guidance scale $[w_{min}, w_{max}]$ | $[3, 5]$ |

# F ADDITIONAL RESULTS

## F.1 RESULTS OF CLASSIFICATION ACCURACY SCORE

Classification Accuracy Score (CAS) (Ravuri & Vinyals, 2019) is proposed to evaluate the utility of the images produced by the generative model for downstream classification tasks. The underlying motivation for this metric is that if the generative model captures the real data distribution, the real data distribution can be replaced by the model-generated data and achieve similar results on downstream tasks like image classification.

Table 6: Comparasion of CAS of different methods on `CIFAR-10` 32×32 dataset.

| Methods | CAS |
|---|---|
| Real | 92.5 |
| *only using the synthetic data.* | |
| ADM | 91.0 |
| ADM-IP | 89.2 |
| ADM-AT (Ours) | **91.6** |
| *using the synthetic data with real data.* | |
| ADM | 95.0 |
| ADM-IP | 94.9 |
| ADM-AT (Ours) | **95.4** |

Following the evaluation pipeline in Ravuri & Vinyals (2019), we train the image classifier in two settings: only on synthetic data or real data augmented with synthetic data, and use the classifier to predict labels on the test set of real data. Synthetic images are generated with a DDIM sampler under 20 NFEs. We use ResNet-18 (He et al., 2016) as the image classifier and train it for 200 epochs with a learning rate of 0.1 and a batch size of 128. We report CAS in the `CIFAR-10` dataset at a resolution of 32×32 in Table 6. The results indicate that our method consistently performs better than other baseline methods on CAS metric in both settings. Although CAS with synthetic data cannot surpass real data, it demonstrates significant potential for enhancing classifier accuracy when employed as an augmentation technique alongside real data.

Table 7: Comparasion of AT with TS-DDIM on `CIFAR10` 32×32. Both models are based on the ADM backbone. The results of TS are taken directly from the original paper.

| Methods \ NFEs | 50 | 20 | 10 | 5 |
|---|---|---|---|---|
| ADM-TS-DDIM | 3.52 | 5.35 | 10.73 | 26.94 |
| ADM-AT (Ours) | **3.07** | **4.40** | **9.30** | **26.38** |

### F.2 COMPARISON TO TS-DDIM

Li et al. (2024) introduces another approach named Time-Shift (TS) to alleviate the DPM distribution mismatch by searching for coupled time steps in sampling. Table 7 shows the comparison between our AT method with TS on `CIFAR-10` with the DDIM Sampler. Both methods are based on the ADM pretrained model (Dhariwal & Nichol, 2021) as a backbone, which is the same as Section 6.2. We observe our method consistently better than the TS method across various sampling steps.

### F.3 RESULTS OF MORE NFES

We present results obtained with various samplers under 100 or 200 NFEs on `CIFAR10` 32x32 and `ImageNet` 64x64 in Table 8 and Table 9, respectively. The results show that our method is still effective for samplers under hundreds of NFEs.

### F.4 RESULTS OF MORE METRICS

We present the results of more generation quality metrics, including sFID, Inception Score (IS), Precision, and Recall, on `CIFAR10` 32x32 (Table 10 and Table 11) and `ImageNet` 64x64 (Table 12 and Table 13). The evaluation is performed following Dhariwal & Nichol (2021). We observe that our method shows effectiveness across these metrics.

Table 8: Sample quality measured by FID ↓ of various sampling methods of DPM under 100 or 200 NFEs on `CIFAR10` 32x32.

| Methods | IDDPM | | DDIM | | ES | | DPM-Solver | |
|---|---|---|---|---|---|---|---|---|
| | 100 | 200 | 100 | 200 | 100 | 200 | 100 | 200 |
| ADM-FT | 3.34 | 3.02 | 4.02 | 4.22 | 2.38 | 2.45 | 2.97 | **2.97** |
| ADM-IP | 2.83 | 2.73 | 6.69 | 8.44 | 2.97 | 3.12 | 10.10 | 10.11 |
| ADM-AT (Ours) | **2.52** | **2.46** | **3.19** | **3.23** | **2.18** | **2.35** | **2.83** | 3.00 |

Table 9: Sample quality measured by FID ↓ of various sampling methods of DPM under 100 or 200 NFEs on `ImageNet` 64x64.

| Methods | IDDPM | | DDIM | | ES | | DPM-Solver | |
|---|---|---|---|---|---|---|---|---|
| | 100 | 200 | 100 | 200 | 100 | 200 | 100 | 200 |
| ADM-FT | 3.88 | 3.48 | 4.71 | 4.38 | 3.07 | **2.98** | **4.20** | 4.13 |
| ADM-IP | 3.55 | **3.08** | 8.53 | 10.43 | 3.36 | 3.31 | 9.75 | 9.77 |
| ADM-AT (Ours) | **3.35** | 3.16 | **4.58** | **4.34** | **3.05** | 3.10 | 4.31 | **4.10** |

Table 10: Comparison of sFID ↓ and IS ↑ on `CIFAR10` 32x32.

(a) IDDPM

| | 5 | | 8 | | 10 | | 20 | | 50 | |
|---|---|---|---|---|---|---|---|---|---|---|
| | sFID | IS | sFID | IS | sFID | IS | sFID | IS | sFID | IS |
| ADM | 20.95 | 8.25 | 25.03 | 8.51 | 23.56 | 8.50 | 16.01 | 9.14 | 6.81 | 9.49 |
| ADM-IP | 25.81 | 7.02 | **24.51** | 8.04 | **19.02** | 8.50 | 8.99 | 9.28 | 5.32 | **9.66** |
| ADM-AT | **19.78** | **8.71** | 25.67 | **8.66** | 23.09 | **8.77** | **6.01** | **9.30** | **5.04** | 9.65 |

(b) DDIM

| | 5 | | 8 | | 10 | | 20 | | 50 | |
|---|---|---|---|---|---|---|---|---|---|---|
| | sFID | IS | sFID | IS | sFID | IS | sFID | IS | sFID | IS |
| ADM | 12.75 | 7.76 | 8.53 | 8.62 | 8.39 | 8.70 | 6.19 | 9.08 | 4.99 | 9.19 |
| ADM-IP | 15.53 | 7.55 | 8.00 | **8.98** | 7.12 | **9.15** | **5.30** | **9.41** | 5.64 | 9.49 |
| ADM-AT | **12.56** | **7.97** | **7.93** | 8.90 | **7.08** | 8.90 | 5.37 | 9.17 | **4.66** | **9.51** |

(c) ES

| | 5 | | 8 | | 10 | | 20 | | 50 | |
|---|---|---|---|---|---|---|---|---|---|---|
| | sFID | IS | sFID | IS | sFID | IS | sFID | IS | sFID | IS |
| ADM | 27.39 | 6.14 | 14.91 | 8.33 | 10.04 | 8.79 | 5.45 | 9.55 | 4.12 | 9.62 |
| ADM-IP | 34.70 | 5.73 | 16.84 | 8.23 | 10.89 | 8.88 | 4.94 | 9.59 | 4.08 | 9.70 |
| ADM-AT | **16.84** | **6.97** | **10.33** | **8.60** | **8.00** | **8.95** | **4.78** | **9.65** | **4.04** | **9.77** |

(d) DPM-Solver

| | 5 | | 8 | | 10 | | 20 | | 50 | |
|---|---|---|---|---|---|---|---|---|---|---|
| | sFID | IS | sFID | IS | sFID | IS | sFID | IS | sFID | IS |
| ADM | 11.82 | 8.00 | 5.79 | 9.12 | **5.05** | 9.41 | **4.43** | 9.78 | **4.32** | 9.82 |
| ADM-IP | 26.46 | 7.09 | 5.93 | 9.19 | 5.49 | 9.45 | 7.53 | 9.66 | 8.37 | 9.75 |
| ADM-AT | **11.19** | **8.43** | **5.10** | **9.35** | 5.29 | **9.65** | 4.75 | **10.03** | 4.59 | **9.93** |

# G  MORE ANALYSIS

## G.1  EFFICIENT AT VS STANDARD AT

In this section, we conduct an ablation of the AT method in diffusion model training. We compare the performance of our used efficient AT and a standard AT method PGD on `CIFAR-10` dataset at the resolution of 32×32. The adversarial step $K$ is set to be 3 for both methods. We fine-tune both

Table 11: Comparison of Precision (P) ↑ and Recall (R) ↑ on `CIFAR10` 32x32.

(a) IDDPM

|  | 5 | | 8 | | 10 | | 20 | | 50 | |
|  | P | R | P | R | P | R | P | R | P | R |
|---|---|---|---|---|---|---|---|---|---|---|
| ADM | **0.54** | **0.47** | **0.59** | **0.45** | 0.61 | **0.46** | 0.64 | 0.52 | 0.68 | 0.58 |
| ADM-IP | **0.54** | 0.39 | **0.59** | 0.43 | 0.61 | **0.46** | 0.66 | 0.54 | 0.68 | 0.59 |
| ADM-AT | 0.52 | **0.47** | 0.57 | **0.45** | **0.62** | **0.46** | **0.68** | **0.55** | **0.69** | **0.59** |

(b) DDIM

|  | 5 | | 8 | | 10 | | 20 | | 50 | |
|  | P | R | P | R | P | R | P | R | P | R |
|---|---|---|---|---|---|---|---|---|---|---|
| ADM | 0.57 | **0.47** | 0.59 | 0.52 | 0.61 | 0.52 | 0.64 | 0.52 | 0.63 | 0.60 |
| ADM-IP | 0.57 | 0.44 | **0.62** | **0.53** | **0.63** | **0.56** | 0.65 | **0.60** | 0.65 | **0.61** |
| ADM-AT | **0.59** | 0.46 | **0.62** | 0.52 | **0.63** | 0.54 | **0.65** | 0.58 | **0.66** | **0.61** |

(c) ES

|  | 5 | | 8 | | 10 | | 20 | | 50 | |
|  | P | R | P | R | P | R | P | R | P | R |
|---|---|---|---|---|---|---|---|---|---|---|
| ADM | 0.54 | 0.37 | 0.60 | 0.48 | 0.61 | 0.52 | 0.64 | 0.52 | 0.63 | 0.60 |
| ADM-IP | 0.46 | 0.32 | 0.58 | 0.45 | 0.62 | 0.51 | **0.67** | **0.58** | **0.68** | 0.60 |
| ADM-AT | **0.61** | **0.45** | **0.64** | **0.51** | **0.65** | **0.54** | 0.65 | **0.58** | 0.66 | **0.61** |

(d) DPM-Solver

|  | 5 | | 8 | | 10 | | 20 | | 50 | |
|  | P | R | P | R | P | R | P | R | P | R |
|---|---|---|---|---|---|---|---|---|---|---|
| ADM | 0.61 | 0.47 | **0.65** | 0.58 | **0.65** | 0.59 | 0.66 | 0.61 | 0.63 | **0.62** |
| ADM-IP | 0.49 | 0.32 | **0.65** | 0.58 | **0.65** | 0.59 | 0.62 | 0.58 | 0.61 | 0.56 |
| ADM-AT | **0.62** | **0.49** | **0.65** | **0.59** | **0.65** | **0.61** | **0.67** | **0.62** | **0.65** | 0.61 |

models from the same pretrained ADM model with 100K update iterations of the model. The results are shown in Table 14. We report the results of 4 sampler settings (method-NFEs): IDDPM-50, DDIM-50, ES-20, and DPM-Solver-10.

We observe that efficient AT achieves performance comparable to or even better than PGD with the same model update iterations while accelerating the training (2.6× speed-up). Thus, we propose applying the efficient AT method for our adversarial training framework.

## G.2 CONVERGENCE OF AT ON DIFFUSION MODELS

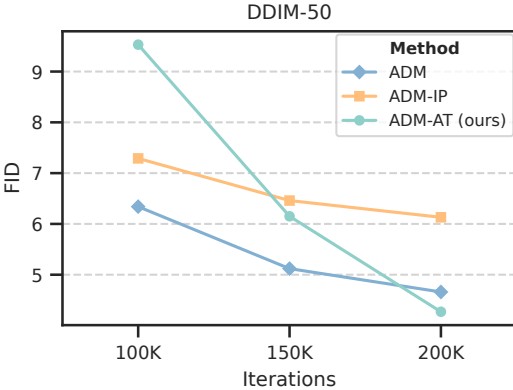

Figure 2: The convergence of methods trained from scratch on `CIFAR-10` 32 × 32. We use the DDIM sampler with 50 NFEs for sampling.

Table 12: Comparison of sFID ↓ and IS ↑ on `ImageNet` 64x64.

(a) IDDPM

|  | 5 | | 8 | | 10 | | 20 | | 50 | |
|  | sFID | IS | sFID | IS | sFID | IS | sFID | IS | sFID | IS |
|---|---|---|---|---|---|---|---|---|---|---|
| ADM | 26.17 | 12.55 | **36.34** | 22.61 | 40.52 | 26.55 | 26.08 | 39.10 | 11.35 | 45.68 |
| ADM-IP | 40.90 | 12.19 | 47.98 | 23.47 | 37.72 | 27.86 | 25.06 | **39.40** | 6.75 | 44.87 |
| ADM-AT | **24.82** | **14.50** | 37.04 | **23.84** | **36.50** | **30.03** | **22.83** | 39.12 | **5.69** | **46.25** |

(b) DDIM

|  | 5 | | 8 | | 10 | | 20 | | 50 | |
|  | sFID | IS | sFID | IS | sFID | IS | sFID | IS | sFID | IS |
|---|---|---|---|---|---|---|---|---|---|---|
| ADM | 27.74 | 14.30 | 14.27 | 25.88 | 12.78 | 28.29 | 8.84 | 33.54 | 6.31 | 38.08 |
| ADM-IP | 52.08 | 10.21 | 16.40 | 22.03 | 11.70 | 25.94 | 9.09 | 32.04 | 15.14 | 31.62 |
| ADM-AT | **25.49** | **14.82** | **10.68** | **26.62** | **9.22** | **29.29** | **6.41** | **34.33** | **4.66** | **39.36** |

(c) ES

|  | 5 | | 8 | | 10 | | 20 | | 50 | |
|  | sFID | IS | sFID | IS | sFID | IS | sFID | IS | sFID | IS |
|---|---|---|---|---|---|---|---|---|---|---|
| ADM | 34.55 | 13.29 | 42.32 | 24.98 | 34.44 | 29.36 | 14.44 | 40.45 | 6.41 | 45.36 |
| ADM-IP | 44.81 | 10.07 | 41.01 | 22.44 | 30.12 | 27.66 | **10.13** | 39.50 | **4.67** | 44.69 |
| ADM-AT | **29.72** | **16.49** | **33.58** | **27.85** | **27.64** | **31.94** | 10.22 | **42.18** | 5.10 | **45.59** |

(d) DPM-Solver

|  | 5 | | 8 | | 10 | | 20 | | 50 | |
|  | sFID | IS | sFID | IS | sFID | IS | sFID | IS | sFID | IS |
|---|---|---|---|---|---|---|---|---|---|---|
| ADM | 25.70 | 24.34 | 11.08 | 34.77 | 8.05 | **37.45** | 5.35 | **40.54** | 4.69 | **41.31** |
| ADM-IP | 42.68 | 16.93 | **7.47** | 33.85 | 7.22 | 33.57 | 14.74 | 31.29 | 18.99 | 30.32 |
| ADM-AT | **20.79** | **26.32** | 7.60 | **34.89** | **6.36** | 36.51 | **4.51** | 38.79 | **4.22** | 39.10 |

Figure 3: The convergence of methods fine-tuned from a same pretrained model on `CIFAR-10` $32 \times 32$. We compare the performance of methods on various samplers.

In classification tasks, adding adversarial perturbations usually slows the convergence of model training (Zhu et al., 2020). We are interested to see whether AT also affects the convergence of the diffusion training process.

Firstly, we explore the convergence of models trained from scratch. We utilize DDIM as the sampler with 50 NFEs and the results are shown in Figure 2. We observe that our AT method and ADM-IP exhibit slower convergence compared to ADM at the beginning (before 100K iterations), while as training more iterations (200K), our AT method shows a notable advantage.

Moreover, we explore the convergence of models under fine-tuning setting and the results are shown in Figure 3. We observe under this setting, when given a pretrained diffusion model like ADM, fine-tuning it with our proposed AT improves performance faster than other baselines. Overall, we observe that incorporating AT with a diffusion framework does not affect the convergence of the model much, especially in the fine-tuning setting.

Table 13: Comparison of Precision (P) ↑ and Recall (R) ↑ on `ImageNet` 64x64.

(a) IDDPM

|  | 5 | | 8 | | 10 | | 20 | | 50 | |
|  | P | R | P | R | P | R | P | R | P | R |
|---|---|---|---|---|---|---|---|---|---|---|
| ADM | 0.34 | 0.48 | 0.46 | **0.50** | 0.51 | 0.48 | 0.65 | 0.52 | 0.73 | 0.57 |
| ADM-IP | 0.39 | 0.39 | **0.50** | 0.45 | **0.56** | 0.48 | 0.68 | **0.55** | 0.73 | **0.60** |
| ADM-AT | **0.40** | **0.50** | **0.50** | **0.50** | 0.55 | **0.49** | **0.69** | 0.52 | **0.77** | 0.59 |

(b) DDIM

|  | 5 | | 8 | | 10 | | 20 | | 50 | |
|  | P | R | P | R | P | R | P | R | P | R |
|---|---|---|---|---|---|---|---|---|---|---|
| ADM | 0.42 | **0.47** | 0.54 | **0.56** | 0.58 | **0.58** | 0.65 | 0.60 | 0.69 | **0.61** |
| ADM-IP | 0.38 | 0.40 | 0.51 | 0.53 | 0.55 | 0.57 | 0.63 | **0.61** | 0.62 | **0.61** |
| ADM-AT | **0.44** | 0.43 | **0.58** | 0.55 | **0.62** | 0.56 | **0.69** | 0.59 | **0.72** | **0.61** |

(c) ES

|  | 5 | | 8 | | 10 | | 20 | | 50 | |
|  | P | R | P | R | P | R | P | R | P | R |
|---|---|---|---|---|---|---|---|---|---|---|
| ADM | 0.40 | 0.44 | 0.52 | 0.47 | 0.58 | 0.48 | 0.69 | 0.55 | 0.73 | 0.59 |
| ADM-IP | 0.37 | 0.35 | 0.49 | 0.44 | 0.56 | **0.49** | 0.68 | **0.57** | 0.72 | **0.60** |
| ADM-AT | **0.44** | **0.46** | **0.58** | **0.48** | **0.63** | **0.49** | **0.73** | 0.55 | **0.76** | 0.59 |

(d) DPM-Solver

|  | 5 | | 8 | | 10 | | 20 | | 50 | |
|  | P | R | P | R | P | R | P | R | P | R |
|---|---|---|---|---|---|---|---|---|---|---|
| ADM | 0.51 | 0.49 | 0.65 | 0.58 | 0.67 | **0.60** | 0.69 | **0.62** | 0.69 | **0.62** |
| ADM-IP | 0.39 | 0.44 | 0.64 | **0.60** | 0.64 | **0.60** | 0.59 | 0.60 | 0.57 | 0.59 |
| ADM-AT | **0.56** | **0.50** | **0.68** | 0.57 | **0.69** | 0.59 | **0.72** | 0.60 | **0.71** | 0.61 |

Table 14: Comparison of different AT methods used in our AT framework. All models are trained with the same model-updating iterations while the efficient AT has less training time.

| Methods | FID | | | | Training Time |
|  | IDDPM-50 | DDIM-50 | ES-20 | DPM-Solver-10 | Speedup |
|---|---|---|---|---|---|
| Standard AT PGD-3 | 4.02 | **3.37** | 6.42 | 7.60 | 1.0× |
| Efficient AT (Ours) | **3.97** | 3.42 | **5.98** | **6.05** | **2.6×** |

Table 15: Comparison of different adversarial learning rate $\alpha$ of our AT framework on `CIFAR10` 32x32. IDDPM is adopted as the inference sampler.

| $\alpha$ \ NFEs | 5 | 8 | 10 | 20 | 50 |
|---|---|---|---|---|---|
| $\alpha = 0.05$ | 51.72 | 32.09 | 25.48 | 10.38 | 4.36 |
| $\alpha = 0.1$ | 37.15 | 23.59 | 15.88 | 6.60 | 3.34 |
| $\alpha = 0.5$ | 63.73 | 40.08 | 27.57 | 7.23 | 3.42 |

Table 16: Comparison of different adversarial learning rate $\alpha$ of our AT framrwork on `ImageNet` 64x64. IDDPM is adopted as the inference sampler.

| $\alpha$ \ NFEs | 5 | 8 | 10 | 20 | 50 |
|---|---|---|---|---|---|
| $\alpha = 0.1$ | 56.92 | 27.39 | 24.06 | 10.17 | 5.82 |
| $\alpha = 0.5$ | 45.65 | 23.79 | 19.18 | 8.28 | 4.01 |
| $\alpha = 0.8$ | 46.92 | 28.46 | 22.47 | 9.70 | 4.25 |

Table 17: Comparison of different perturbation norms ($l_1$, $l_2$ $l_\infty$) of our AT framework on `CIFAR10` 32x32.

| Perturbation Norm | IDDPM-50 | DDIM-50 | ES-20 | DPM-Solver-10 |
|---|---|---|---|---|
| $l_1$ | 4.45 | 4.91 | 4.72 | 5.05 |
| $l_2$ | 3.34 | 3.07 | 4.36 | 4.81 |
| $l_\infty$ | 3.87 | 3.63 | 4.48 | 5.32 |

### G.3 MORE ABLATION STUDY

**Ablation on $\alpha$**    We investigate the impact of adversarial learning rate $\alpha$ in our framework. The results of various $\alpha$ on `CIFAR10` 32x32 and `ImageNet` 64x64 are shown in Table 15 and Table 16, respectively. We observe that $\alpha$ set to 0.1 is better on `CIFAR10` 32x32 and $\alpha = 0.5$ is better for `ImageNet` 64x64. That says, the image in larger size corresponds to larger optimal perturbation level $\alpha$. We speculate this is because we use the perturbation measured under $\ell_2$-norm, where the $\ell_2$-norm of vector will increase with its dimension.

**Ablation on perturbation norm**    During our experiments, we adopt $\ell_2$-adversarial perturbation. Actually, perturbations in Euclidean space under different $\ell_p$ norm are equivalent with each other, e.g., for vector $\boldsymbol{\delta} \in \mathbb{R}^d$, it holds $\|\boldsymbol{\delta}\|_\infty \leq \|\boldsymbol{\delta}\|_2 \leq \sqrt{d}\|\boldsymbol{\delta}\|_\infty$. Therefore, we select $\|\cdot\|_2$ as representation in our paper. Next, we explore the proposed ADM-AT under different adversarial perturbations.

The results are in Table 17. We found that our method under $\ell_2$-perturbation is more stable and indeed has better performance, thus we suggest to use $\ell_2$-perturbation as in the main body of this paper.

### G.4 QUALITATIVE COMPARISONS

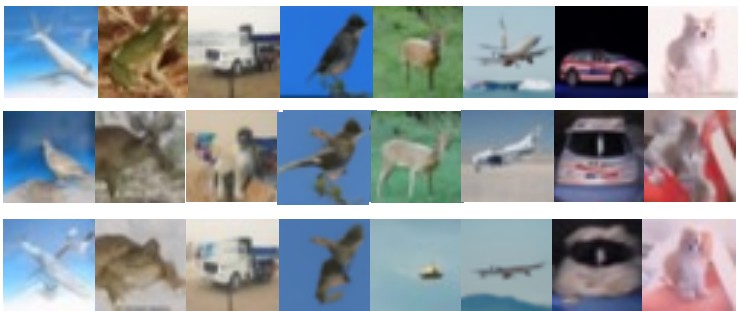

Figure 4: The qualitative comparsions of ADM-AT (top, FID 6.60), ADM-IP (middle, FID 7.81), and ADM (bottom, FID 10.58) on `CIFAR10` $32 \times 32$. We use the IDDPM sampler with 20 NFEs for sampling.

Figure 4, 5, 6, 7 show the qualitative comparisons between our proposed AT method and baselines. Our proposed AT method generates more realistic and higher-fidelity samples. We attribute this to our AT algorithm mitigates the distribution mismatch problem.

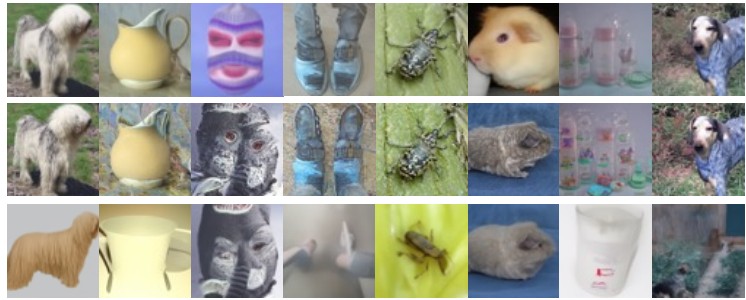

Figure 5: The qualitative comparsions of ADM-AT (top, FID 6.20), ADM-IP (middle, FID 8.40) and ADM (bottom, FID 8.32) on `ImageNet` $64 \times 64$. We use the DDIM sampler with 20 NFEs for sampling.

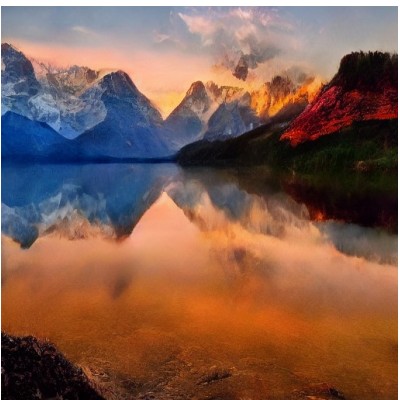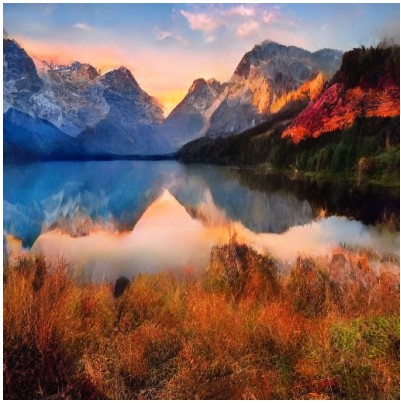

Figure 6: The qualitative comparsions of LCM (left) and LCM-AT (right) with one-step generation. The text prompt is *A photo of beautiful mountain with realistic sunset and blue lake, highly detailed, masterpiece.*

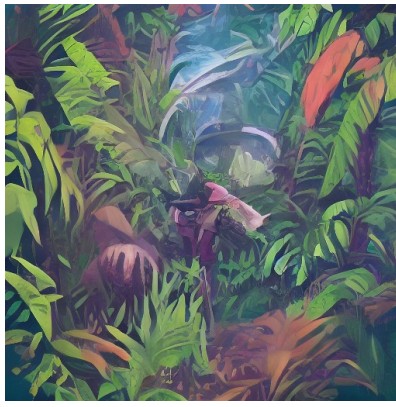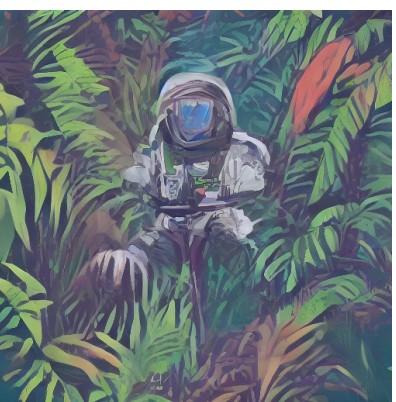

Figure 7: The qualitative comparsions of LCM (left) and LCM-AT (right) with one-step generation. The text prompt is *Astronaut in a jungle, cold color palette, muted colors, detailed, 8k.*

