# OpenReview forum: "Improved Diffusion-based Generative Model with Better Adversarial Robustness"
_ICLR.cc/2025/Conference — ICLR 2025 Poster_

### Official Review · Reviewer_gZkr · 2024-10-28

**Soundness:** 3
**Presentation:** 2
**Contribution:** 3
**Rating:** 6
**Confidence:** 4

**Summary:**

This paper points out the distribution mismatching problem in traditional training of diffusion-based models (DPM) and proposes to conduct efficient adversarial training (AT) during the training of DPM to mitigate this problem. Theoretical analysis is strong enough to support its argument and experiments also verify the effectiveness of the proposed method.

**Strengths:**

1. The motivation for mitigating distribution mismatching is clear and important for efficient sampling.

2. This paper provides strong theoretical support for implementing adversarial training to correct distribution mismatching, making this method convincing.

**Weaknesses:**

1. The experimental results may not be enough, for example, for Table 1 and Table 2, more NFEs should also be verified, although this method can improve efficient sampling, whether is adaptable and robust for more denoising steps should also be verified.

2. Some complex derivations in supplementary material are too brief to understand, such as Eq(30) and Eq(59-62), I'm not sure if there are any typos in them, I suggest checking the equations carefully and modifying them.

**Questions:**

1.  As the weakness above, for Table 1 and Table 2, more NFEs should also be verified.

2. Why not also try generation using consistency models on benchmark datasets such as CIFAR10 and ImageNet, which can be more common and convincing?

3. Derivations in supplementary material should be checked carefully and written with more details.

4. Why efficient AT can improve performance compared with PGD is a bit confusing. Intuitively, PGD should be more accurate to find $\delta_t$, thus more deep insights should be provided here.

---

> ### Author Response · Authors · 2024-11-22
>
> Thanks for your valuable comments and suggestions. Here we address your concerns as follows.
>
> **Q1**: More NFEs should also be verified, although this method can improve efficient sampling, whether is adaptable and robust for more denoising steps should also be verified.
>
> **A1**: Following your suggestion, we report the results with more NFEs (100, 200) below and add them in Appendix F.3 of the revised paper.
>
> Table 5. Sample quality measured by FID $\downarrow$ of various sampling methods of DPM under 100 or 200 NFEs on $\texttt{CIFAR10}$ 32x32.
> |Methods-NFEs |IDDPM-100|IDDPM-200|DDIM-100|DDIM-200|ES-100|ES-200|DPM-Solver-100|DPM-Solver-200|
> |-|-|-|-|-|-|-|-|-|
> |ADM-FT | 3.34 | 3.02 | 4.02 | 4.22 | 2.38 | 2.45 | 2.97 | **2.97**|
> |ADM-IP | 2.83 | 2.73 | 6.69 | 8.44 | 2.97 | 3.12 | 10.10 | 10.11|
> |ADM-AT (Ours) | **2.52** | **2.46** | **3.19** | **3.23** | **2.18** | **2.35** | **2.83** | 3.00|
>
> Table 6. Sample quality measured by FID $\downarrow$ of various sampling methods of DPM under 100 or 200 NFEs on $\texttt{ImageNet}$ 64x64.
> |Methods-NFEs |IDDPM-100|IDDPM-200|DDIM-100|DDIM-200|ES-100|ES-200|DPM-Solver-100|DPM-Solver-200|
> |-|-|-|-|-|-|-|-|-|
> |ADM-FT | 3.88 | 3.48 | 4.71 | 4.38 | 3.07 | **2.98** | **4.20** | 4.13 |
> |ADM-IP | 3.55 | **3.08** | 8.53 | 10.43 | 3.36 | 3.31 | 9.75 | 9.77 |
> |ADM-AT (Ours) | **3.35** | 3.16 | **4.58** | **4.34** | **3.05** | 3.10 | 4.31 | **4.10** |
>
> As can be seen, our method is still effective with hundreds of NFEs.
>
>
>
>
> **Q2**: Some complex derivations in supplementary material are too brief to understand, such as Eq(30) and Eq(59-62), I'm not sure if there are any typos in them, I suggest checking the equations carefully and modifying them.
>
> **A2**: Thanks for pointing out this, for these complex equations, we have revised them to be more clear in the revised version. Please check them accordingly.
>
> **Q3**: Consistency models on benchmark datasets such as CIFAR10 and ImageNet, which can be more common and convincing?
>
> **A3**: Following your suggestion, we conduct the consistency model experiments on ImageNet 64x64.
> Note that since the limited computational resources, we can't directly use the hyperparameters as [1], instead we train the models for 300K iterations with a batch size of 512. Compared with the baseline method CM, our proposed AT method improves the one-step FID from 7.80 to 7.23. We will continue the training process and revise them into the final version.
>
> **Q4**: Derivations in supplementary material should be checked carefully and written with more details.
>
> **A4**: Thank you for your advice, we have carefully checked the derivations in the revised version, and made them more readable.
>
> **Q5** Why efficient AT can improve performance compared with PGD is a bit confusing.
>
> **A5**: As found in [2], for adversarial-type training, in classification, AT takes a balance between accuracy and robustness, i.e., obtaining robustness may sacrifice accuracy. We speculate this phenomenon also holds in the diffusion model, i.e., too strong perturbation may sacrifice the noise prediction accuracy. That explains why strong PGD has slightly worse performance than our efficient AT in some situations.
>
> Reference
>
> [1] Consistency Models. Song et al., 2023.

---

> > ### Comment · Reviewer_gZkr · 2024-11-27
> >
> > I thank the authors for their response and efforts. I will maintain my positive rating for this paper.

---

### Official Review · Reviewer_eJv3 · 2024-10-30

**Soundness:** 3
**Presentation:** 4
**Contribution:** 3
**Rating:** 6
**Confidence:** 4

**Summary:**

This paper identifies the distribution mismatch problem in the training and sampling processes. Consequently, they propose a distributionally robust optimization procedure in the training to bridge the gap. The authors apply the method to both diffusion models and the consistent model, and demonstrate the effectiveness of the proposed method on several benchmarks.

**Strengths:**

1. Identifying and formulating the distribution mismatch problem in diffusion model is an important problem in practice.
2. The proposed solution is elegant, supported by sufficient theoretical analysis. The derivations of the solution is clear and sound.
3. The writing is fairly clear.

**Weaknesses:**

My main concern on this paper is the evaluation. Currently the proposed method is only evaluated using the ADM model. I wonder whether the effectiveness on more advanced model such as the stable diffusion still holds?

Furthermore, the authors only use FID score as the evaluation metric, while it is easy to evaluate the results using other metrics such as IS, sFID, precision, recall, as done in the ADM paper. Why these metrics are not included?

**Questions:**

The paper is a good one in general. I like how the problem is formulated and how the solution is derived. However, given the current evaluation (see the weakness), I am not fully convinced the proposed method is an effective way to deal with the problem. I would like to see how the authors respond to my concerns.

---

> ### Author Response · Authors · 2024-11-22
>
> Thanks for your valuable comments and suggestions. Here we address your concerns as follows.
>
> **Q1**: My main concern in this paper is the evaluation. Currently, the proposed method is only evaluated using the ADM model. I wonder whether the effectiveness of **more advanced model** such as the **stable diffusion** still holds?
>
> **A1**: Thanks for your valuable suggestion. We apply our method to LCM under Stable Diffusion v1.5 in Section 6.3, and the experimental results show the effectiveness of our method. To further address your concern, we finetune stable diffusion v1.5 under the framework of DPM by our proposed adversarial training. The evaluation results are summarized below.
>
> Table 5. Comparison of FID on MS-COCO dataset with DDIM sampler.
> | |5 Steps|10 Steps|
> |-|-|-|
> |SD v1.5 | 32.95 | 18.99|
> |SD v1.5-AT |25.82|15.65|
>
> As can be seen, our method still has better performance, compared with the baseline method.
>
> **Q2**:  The authors only use FID score as the evaluation metric, while it is easy to evaluate the results using other metrics such as IS, sFID, precision, recall, as done in the ADM paper. Why these metrics are not included?
>
> **A2**:
> Thanks for your suggestion. We list the IS, sFID, Precision, and Recall of CIFAR10 32x32 with DDIM sampler as a representation as below.
> The results of our ADM-AT also outperform the baseline across metrics overall.
> More results of various samplers and datasets can be found in Appendix F.4 of the revised paper.
>
> Table 5. Comparison of sFID $\downarrow$ and IS $\uparrow$ on $\texttt{CIFAR10}$ 32x32.
> |NFE-metric |5-sFID|5-IS|8-sFID|8-IS|10-sFID|10-IS|20-sFID|20-IS|50-sFID|50-IS|
> |-|-|-|-|-|-|-|-|-|-|-|
> |ADM | 12.75 | 7.76 | 8.53 | 8.62 | 8.39 | 8.70 | 6.19 | 9.08 | 4.99 | 9.19|
> |ADM-AT | **12.56** | **7.97** | **7.93** | **8.90** | **7.08** | **8.90** | **5.37** | **9.17** | **4.66** | **9.51**|
>
> Table 6. Comparison of Precision (P) $\uparrow$ and Recall (R) $\uparrow$ on $\texttt{CIFAR10}$ 32x32.
> |NFE-metric |5-Precision|5-Recall|8-Precision|8-Recall|10-Precision|10-Recall|20-Precision|20-Recall|50-Precision|50-Recall|
> |-|-|-|-|-|-|-|-|-|-|-|
> |ADM | 0.57 |  **0.47** | 0.59 | 0.52 | 0.61 | 0.52 | 0.64 | 0.52 | 0.63 | 0.60|
> |ADM-AT | **0.59** | 0.46 |  **0.62** | 0.52|  **0.63** | **0.54** |  **0.65** | **0.58** |  **0.66** |  **0.61**|

---

> > ### Comment · Reviewer_eJv3 · 2024-11-24
> >
> > I thank the authors for the rebuttal. The additional experiments resolve most of my concerns, thus I am raising my score to 6. I tend to not raising more because I think the result improvement is very marginal, thus it is unclear whether the proposed method would have practical value.

---

> > > ### Author Response · Authors · 2024-11-24
> > >
> > > Dear Reviewer eJv3,
> > >
> > > Thank you for your prompt response. We sincerely appreciate your willingness to consider upgrading the score of our work. We are glad that our reply addressed your concerns, and we welcome any further comments you may have.
> > >
> > > We apologize for any inconvenience, but it seems the score has not yet been adjusted. At your convenience, we would greatly appreciate it if you could update the score.
> > >
> > > Thank you once again for your constructive comments and suggestions.
> > >
> > > Best regards,
> > >
> > > Authors.

---

### Official Review · Reviewer_ouMQ · 2024-11-04

**Soundness:** 3
**Presentation:** 3
**Contribution:** 3
**Rating:** 6
**Confidence:** 3

**Summary:**

This paper studies the training of unconditional diffusion model. In particular, in order to achieve a better generation quality and enable robust learning of the score network, this paper develops a DRO-based method, and prove the DRO objective in training diffusion models can be formulated as an adversarial learning problem. The paper also identifies a similar mismatch issue in the recently proposed consistency model (CM) and demonstrates that AT can address this problem as well. The authors propose efficient AT for both DPM and CM, with empirical studies confirming the effectiveness of AT in enhancing diffusion-based models.

**Strengths:**

1. This paper performs a theoretical analysis of diffusion models and identifies the distribution mismatch problem.

2. This paper further builds a connection between the distribution robust optimization and adversarial learning for diffusion models, and develops an adversarial training method for diffusion models.

3. This paper conducts efficient adversarial training methods on both diffusion models and consistency models in many tasks. Experimental results demonstrate the effectiveness of the developed algorithms.

**Weaknesses:**

1. In general, the algorithm developed in this paper is motivated by the distribution mismatch along the diffusion path. However, there is no experimental results to justify the motivation, there are also no experimental results to verify that the DRO framework can indeed help mitigate the distribution mismatch problem.

2. Proposition 2 has already been discovered in existing theoretical papers [1], see their section 3.1. The authors should comment on this point around Proposition 2.

3. The advantage of ADM-AT is not that significant compared with the ADM method, a more detailed ablation study or theoretical analysis on using adversarial noise or random Gaussian noise should be added.

4. Some statements are not clearly presented. For instance, the description of ADM is not given, the norm notations $\|\|$ are abused, should that be $\ell_1$, $\ell_2$, or $\ell_\infty$?


[1] Chen, Lee, and Lu, Improved Analysis of Score-based Generative Modeling: User-Friendly Bounds under Minimal Smoothness Assumptions, ICML 2023

**Questions:**

1. some ablation studies for different perturbation levels $\alpha$ should be given.
2. Some discussions about different perturbation methods ($\ell_1$, $\ell_2$, or $\ell_\infty$) should be discussed.

---

> ### Author Response · Authors · 2024-11-22
> **Response Part 1**
>
> Thanks for your valuable comments and suggestions. Here we address your concerns as follows.
>
> **Q1**: In general, the algorithm developed in this paper is motivated by the distribution mismatch along the diffusion path. However, there is no experimental results to justify the motivation, there are also no experimental results to **verify that the DRO framework can indeed help mitigate the distribution mismatch problem**.
>
> **A1**: Thanks for your valuable comment, we follow your suggestion to verify the quality of the intermediate $x_{t}$, as the mismatching problem is revealed by this. Concretely, we evaluate the FID between the generated $x_{t}$ by our and baseline methods, and compare them with the ground-truth ones.
> We adopt the IDDPM sampler under 200 NFEs and report the last 10 steps FID-10K in Table 1. We can observe that the ADM-AT achieves better FID on these steps.
>
> Table 1. Comparison of FID of intermediate $x_t$.
> | Step Index|200|199 |198|197|196|195|194|193|192|191|190|
> |-|-|-|-|-|-|-|-|-|-|-|-|
> |ADM|4.94|5.34|11.63|19.61|27.60|35.23|42.37|49.08|55.33|61.09|66.80|
> |ADM-IP|5.23|5.62|12.04|19.69|27.48|34.95|41.89|48.43|54.60 | 60.44|66.04|
> |ADM-AT|**4.52**|**5.00**|**11.37**|**18.90**|**25.38**|**32.01**|**39.83**|**46.05**|**51.87**|**57.44**|**62.73**|
>
> Note that the Step 200 is the endpoint of the sampling process.
>
> **Q2**: Proposition 2 has already been discovered in existing theoretical papers [1], see their section 3.1. The authors should comment on this point around Proposition 2.
>
> **A2**: Thank you for pointing out the important reference. The results in section 3.1 are similar to our Proposition 2, as both of us quantify the KL divergence between the generated samples and the ground-truth ones. However, their results focus on the generated target data $x_{0}$, while ours focuses on all $x_{t}$ ($0\leq t\leq T$), though our techniques are similar. We have mentioned this comparison in the revised version.
>
> **Q3**: The advantage of ADM-AT is not that significant compared with the ADM method, a more detailed ablation study or theoretical analysis on using adversarial noise or random Gaussian noise should be added.
>
> **A3**:
> Our ADM-AT significantly outperforms the baselines ADM or ADM-IP across various samplers, especially under the setting of fewer NFEs (more practical and efficient). As shown in Table 1 of our paper: on CIFAR-10 32x32, with the IDDPM sampler, ADM-AT improves FID from **10.52 to 6.60**. With the DDIM sampler, ADM-AT improves FID from **11.66 to 9.30** under 10 NFEs. With the DPM-Solver, ADM-AT improves FID from **8.00 to 5.84**. These improvements are recognized as significant and practical.
>
> Firstly, the proposed adversarial noise is induced by our theoretical framework under Distributional Robust Optimization to avoid distribution mismatching. As for the comparison with Gaussian noise, the baseline method ADM-IP adds Gaussian noise during training. According to our empirical results, our proposed adversarial noise is significantly better than the Gaussian noise.

---

> ### Author Response · Authors · 2024-11-22
> **Response Part 2**
>
> **Q4**: Some statements are not clearly presented. For instance, the description of ADM is not given.
>
> **A4**: For the description of ADM, it is a UNet-type [2] (with self-attention layer) neural network proposed by [3]. It is a standard architecture for diffusion model under image generation task. We have added this description in the revised version.
>
> **Q5**: The norm $\|\cdot\|$ notations are abused, should that be $\ell_1, \ell_2$ or $\ell_\infty$.
>
> **A5**: Without specifying, we use $\|\cdot\|$ to denote $\ell_{2}$-norm. We have added this description in line 137 of the revised version.
>
>
>
> **Q5**. Some ablation studies for different perturbation levels $\alpha$ should be given.
>
> **A5**:
> We explore the proposed under different perturbation level $\alpha$ on $\texttt{CIFAR10}$ 32x32 and $\texttt{ImageNet}$ 64x64 below. IDDPM is adopted as the inference sampler.
>
> Table 2. Comparison of different adversarial learning rate α on $\texttt{CIFAR10}$ 32x32.
> | $\alpha$ $\backslash$ NFEs|5 | 8 |10 | 20 | 50|
> |-|-|-|-|-|-|
> |$\alpha=0.05$ | 51.72 | 32.09 | 25.48 | 10.38 | 4.36 |
> |$\alpha=0.1$ | **37.15** | **23.59**  | **15.88** | **6.60** | **3.34** |
> |$\alpha=0.5$ | 63.73 | 40.08 | 27.57 | 7.23 | 3.42 |
>
> Table 3. Comparison of different adversarial learning rate $\alpha$ of our AT framrwork on $\texttt{ImageNet}$ 64x64.
> | $\alpha$ $\backslash$ NFEs|5 | 8 |10 | 20 | 50|
> |-|-|-|-|-|-|
> |$\alpha=0.1$ | 56.92 | 27.39 | 24.06  | 10.17 | 5.82|
> |$\alpha=0.5$ | **45.65** | **23.79** | **19.18** | **8.28** | **4.01**|
> |$\alpha=0.8$ | 46.92 | 28.46 | 22.47 | 9.70 | 4.25|
>
> We observe that $\alpha = 0.1$ is better on $\texttt{CIFAR10}$ 32x32 and $\alpha=0.5$ is better for $\texttt{ImageNet}$ 64x64. We observe that image in larger size corresponds to larger optimal perturbation level $\alpha$. We speculate this is because we use the perturbation measured under $\ell_{2}$-norm, where the $\ell_{2}$-norm of vector will increase with its dimension. We have added this ablation study in Appendix G.3 of the revised paper, please check it.
>
>
> **Q6**: Some discussions about different perturbation methods ($\ell_1, \ell_2$ or $\ell_\infty$) should be discussed.
>
> **A6**: These three perturbations are actually mathematically equivalent e.g., for vector $x\in \mathbb{R}^{d}$, it holds $\|x\|_{\infty}\leq \|x\|_{2} \leq \sqrt{d}\|x\|_{\infty}$. Therefore, we select $\|\cdot\|_{2}$ as representation in our paper. To further address your concern, we explore our method under three different perturbation methods ($\ell_1, \ell_2$ or $\ell_\infty$). The results are summarized below.
>
> Table 4. Comparison of different perturbation norms on $\texttt{CIFAR10}$ 32x32.
> |Perturbation Norm |IDDPM-50|DDIM-50|ES-20|DPM-Solver-10|
> |-|-|-|-|-|
> |$l_1$|4.45|4.91|4.72|5.05|
> |$l_2$|**3.34**|**3.07**|**4.36**|**4.81**|
> |$l_{\infty}$|3.87|3.63|4.48|5.32|
>
> During our experiments, we found that our method under $\ell_2$-perturbation is more stable and indeed has better performance, thus we suggest to use $\ell_2$-perturbation as in our paper. This discussion has been added in the Appendix G.3 of the revised paper.
>
>
>
> Reference
>
> [1] Improved Analysis of Score-based Generative Modeling: User-Friendly Bounds under Minimal Smoothness Assumptions. Chen et al., 2023.
>
> [2] Convolutional Networks for Biomedical Image Segmentation. Ronneberger et al., 2015.
>
> [3] Diffusion Models Beat GANs on Image Synthesis. Dhariwal et al., 2021.

---

> > ### Comment · Reviewer_ouMQ · 2024-11-25
> >
> > I thank the authors for their detailed response. I will maintain the positive rating for this paper.

---

### Official Review · Reviewer_tM1A · 2024-11-04

**Soundness:** 3
**Presentation:** 3
**Contribution:** 3
**Rating:** 6
**Confidence:** 4

**Summary:**

The paper proposes to introduce DRO to address the distribution matching problem at training diffusion model.

**Strengths:**

1. The paper present theories to show that DRO can help address the distribution matching problem in training and testing diffusion models.

2. The improvement over baselines on Cifar and Imagenet64 show that DRO is useful.

**Weaknesses:**

1. There is no qualitative comparisons. Authors mainly conduct experiments on Cifar, ImageNet and Laion dataset. It would be better to put some images for more direct comparisons. In addition, the code is not provided.

2. The efficiency comparison. I am wondering how much overhead it brings to adopt eq 14 instead of the classical denoising objective.  I am expecting that it is quite large.

I am giving score of 6 based on the prerequisite that above two concerns are answered during rebuttal.

**Questions:**

as above

---

> ### Author Response · Authors · 2024-11-22
>
> Thanks for your valuable comments and suggestions. Here we address your concerns as follows.
>
> **Q1**: There is no qualitative comparisons. It would be better to put some images for more direct comparisons.
>
> **A1**:
> Thanks for your suggestions. The revised paper adds the qualitative comparisons between ADM-AT with baseline ADM/ADM-IP on CIFAR10 32x32 ( Figure 4)and ImageNet 64x64 (Figure 5). We also add comparisons between LCM-AT and LCM (Figures 6 and 7).
> Overall, models trained with our proposed AT method demonstrate superior performance in generating images for both class-conditional image generation and text-to-image generation tasks.
> Our  AT method generates more realistic and higher-fidelity samples, which is consistent with the results in Tables 1, 2, and 3 in our paper.
>
> **Q2**: In addition, the code is not provided.
>
> **A2**: We will release our code, but currently the code for the consistency model and ImageNet 64x64 is still polished. We provide the diffusion model adversarial training code on CIFAR10 32x32 in the following anonymous link: [code](https://anonymfile.com/89rDY/adm-at.zip).
>
> **Q2**: The efficiency comparison with AT and the classical denoising objective.
>
> **A2**: Yes, as you mentioned, solving the proposed adversarial objective (14) with standard adversarial training technique (e.g., PGD in [1]) is computationally expensive, as we have explored in Table 8 of Appendix G.1.  However, as mentioned in Section 6.1, **we refer to efficient adversarial training [2] algorithm (Algorithm 1) to resolve this**. By doing so, every update of the model under the proposed (14) has a **similar computational cost** to the classical objective. For example, we evaluate 10K updates of the model under methods: ADM/ADM-IP/ADM-AT(Ours). The spending time is 960/961s/970s, respectively.
>
> Reference
>
> [1] Towards deep learning models resistant to adversarial attacks. Madry et al., 2018.
>
> [2] Adversarial training for free! Shafahi et al., 2019.

---

### Author Response · Authors · 2024-11-22
**General Response**

We sincerely thank all the reviewers for their thorough evaluations and valuable constructive feedback. We are encouraged that reviewers find that our paper identifies and formulates the distribution mismatch problem in the diffusion model and builds a connection between the distribution robust optimization and adversarial learning for diffusion models (Reviewer tM1A, ouMQ), provides strong theoretical support for implementing adversarial training (Reviewer gZkr, eJv3, ouMQ), and experimental results demonstrate the effectiveness both on diffusion models and consistency models (Reviewer tM1A, ouMQ).

We have updated our paper to incorporate suggestions on clarifications and experimental results as follows:

(1) Regarding the suggestion of Reviewer eJv3 and gZkr, we have clarified the notation more rigorously and made the derivation more detailed.

(2) On experimental results, we incorporate more evaluation metrics (IS/sFID/Precision/Recall), more models like SD, and evaluate models on more NFEs.

(3) More detailed ablation studies of our AT framework are conducted, including different perturbation methods ($\ell_1, \ell_2$ or $\ell_\infty$) and adversarial learning rates $\alpha$.

(4) We also visualize the generation results for qualitative comparisons between our method and baselines.

---

### Meta-Review · Area_Chair_JWwj · 2024-12-22

**Metareview:**

This paper proposes to use robustness-driven adversarial training to solve the distribution mismatch problem in diffusion models. Additionally, the same idea is applied to distillation methods such as the consistency model to further improve the performance. The proposed method is validated on a series of experimental evaluations. The reviewers had some concerns on the evaluation metrics and ablation studies, which were addressed in the author rebuttal. All reviewers are inclined toward acceptance, a sentiment with which I agree.

**Additional Comments On Reviewer Discussion:**

Before rebuttal, all the reviewers except eJv3 showed a positive view of this paper. The main concerns are on the evaluation metrics in the experiments. The authors added the suggested metrics and showed consistently good results. Reviewer eJv3 increased their rating to 6. The other concerns are mainly about ablation studies and more visualization results. The authors did a good job in answering these questions. My decision is mainly based on the positive rating of the reviewers and also the solid rebuttal from the authors.

---

### Decision · Program_Chairs · 2025-01-22

Accept (Poster)